

# 1 A 14.5 million-year record of East Antarctic Ice Sheet 2 fluctuations from the central Transantarctic Mountains, 3 constrained with cosmogenic ³He, ¹⁰Be, ²¹Ne, and ²⁶Al

Allie Balter[1,2], Gordon Bromley[2,3], Greg Balco[4], Holly Thomas[1], Margaret S. Jackson[3]
[1]School of Earth and Climate Sciences, University of Maine, Orono, Maine, USA
[2]Climate Change Institute, University of Maine, Orono, Maine, USA
[3]Geography, National University of Ireland, Galway, Ireland
[4]Berkeley Geochronology Center, Berkeley, California, USA
*Correspondence to: Allie Balter (abalter@ldeo.columbia.edu)*
**Abstract.** The distribution of moraines in the Transantarctic Mountains affords direct constraint of past ice-marginal
positions of the East Antarctic Ice Sheet (EAIS). Here, we describe glacial-geologic observations and cosmogenic-
nuclide exposure ages from Roberts Massif, an ice-free area in the central Transantarctic Mountains. We measured
cosmogenic ³He, ¹⁰Be, ²¹Ne, and ²⁶Al in 180 dolerite and sandstone boulders collected from 24 distinct deposits. Our
data show that a cold-based EAIS was present, in a configuration similar to today, for many periods over the last ~14.5
Myr, including the mid-Miocene, Late Pliocene, and early-to-mid Pleistocene. Moraine ages at Roberts Massif
increase with distance from, and elevation above the modern ice margin, which is consistent with a persistent EAIS
extent during glacial maxima, and slow, isostatic uplift of the massif itself in response to trough incision by outlet
glaciers. We also employ the exceptionally high cosmogenic-nuclide concentrations in several boulders, along with
multi-isotope measurements in sandstone boulders, to infer extremely low erosion rates (<< 5 cm/Myr) over the period
covered by our record. Although our data are not a direct measure of ice volume, the Roberts Massif glacial record
indicates that the EAIS was present and similar to its current configuration during at least some periods when global
temperature was believed to be warmer and/or atmospheric $CO_2$ concentrations were likely higher than today.
**1 Introduction**
In this paper, we describe glacial deposits preserved in the central Transantarctic Mountains (TAM, Figure 1) that
provide unambiguous evidence for the presence of the East Antarctic Ice Sheet (EAIS), in a configuration similar to
today, for periods of the middle Miocene, late Pliocene, and early to middle Pleistocene. Our chronology therefore
provides geologic targets for ice volume reconstructions derived from marine proxy records and sea-level estimates.
Current estimates of pre-Pleistocene EAIS ice volume are based largely on $\delta^{18}O$ of benthic foraminifera (e.g.,
Shevenell et al., 2008), which primarily records global temperature and ice volume, and farfield sea-level indicators
(e.g., Miller et al., 2005), such as raised shorelines (e.g., Rovere et al., 2014). These proxy records (e.g., Holbourn et
al., 2013), along with stratigraphic evidence from ice proximal sediment cores (Levy et al., 2016) and modeling studies
(Gasson et al., 2016), suggest that during the middle Miocene the EAIS oscillated between states both larger and



smaller than present in response to fluctuations in $CO_2$ and temperature. After ~14 Ma, such proxy records suggest
general presence of the EAIS, but with potentially significant retreat during past warm periods, such as the mid-
Pliocene Warm Period (3.3–3.0 Ma) (e.g., Dutton et al., 2015 and references therein), when temperatures are thought
to have been 2–3°C warmer than preindustrial (Haywood et al., 2013) and $CO_2$ was ~400 ppm (Pagani et al., 2010;
Seki et al., 2010). Although valuable for elucidating long-term trends in sea-level change, these proxy records do not
directly record the volume of specific ice sheets. In contrast, glacial deposits from ice-free areas of Antarctica itself
provide direct geologic evidence for past ice sheet variability.
Previous geomorphic and glacial chronologic studies in the Transantarctic Mountains (TAM), a ~3000 km-long
topographic barrier through which outlet glaciers of the EAIS drain into the Ross Sea Embayment (Figure 1), suggest
the presence of pre-Pleistocene glacial deposits. Two distinct categories of deposits characterize the Antarctic glacial-
geologic record: basal tills of the Sirius Group (e.g., Mayewski, 1975; Mercer, 1972), which indicate at least one
period of temperate glaciation, and thin, bouldery drifts and moraines deposited by ice frozen to the bed (e.g., Prentice
et al., 1986), which overlie the older temperate deposits. In southern Victoria Land, Schaefer et al., (1999) reported a
minimum age of > 10 Ma for Sirius Group tills at Mt. Fleming. Similarly, relict subglacial flood deposits in the
Coombs Hills resulting from wet-based glaciation afford [3]He ages between ~8.5 and 10.5 Ma, assuming zero erosion,
and as much as ~15 Ma if erosion rates of 0.03–0.06 m/Ma are applied (Margerison et al., 2005). In the same region,
[40]Ar/[39]Ar ages on in situ ash layers interbedded with cold-based ablation tills in the Asgard Range date the transition
from temperate to polar glaciation to between 15 and 13.6 Ma (Sugden and Denton, 2004). The preservation of such
deposits over the last ~15 Ma has been invoked as evidence for persistent polar desert conditions, and by extension
the presence of the EAIS, since that time (Denton et al., 1993).
Chronologic constraints on the overlying cold-based deposits come primarily from surface-exposure dating, which
has been employed at several locations throughout the TAM, including southern Victoria Land (Brook et al., 1995,
1993; Brown et al., 1991; Bruno et al., 1997; Ivy-Ochs et al., 1995; Strasky et al., 2009); Beardmore (Ackert and Kurz,
2004) and Law (Kaplan et al., 2017) Glaciers in the central TAM; and Scott (Spector et al., 2017) and Reedy (Bromley
et al., 2010; Todd et al., 2010) Glaciers in the southern TAM. Approximately 30 previously published exposure ages
(see ICE-D:ANTARCTICA online archive: http://antarctica.ice-d.org) indicate the preservation of cold-based glacial
landforms in Antarctica that are at least 5 Ma in age. For example, a prominent boulder moraine in the Dominion
Range, upper Beardmore Glacier, was dated with [3]He to 5.2 Ma (Ackert and Kurz, 2004). Similarly, [10]Be ages from
erratic boulders at Reedy Glacier suggest deposition of the 'Reedy E drift' at > ~5 Ma (Bromley et al., 2010).



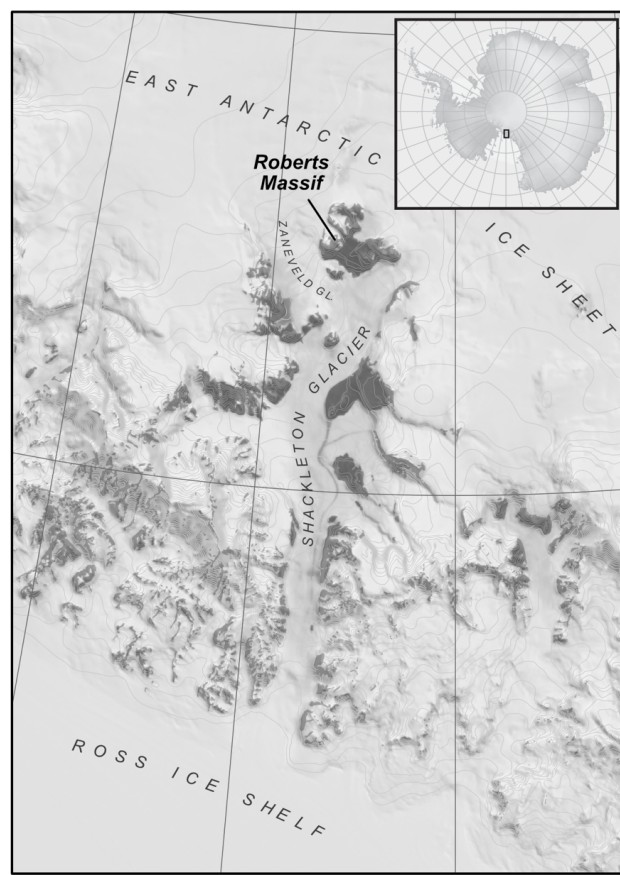

**Figure 1.** Location of Roberts Massif. The massif lies at the head of the Shackleton Glacier, which flows from the polar plateau of the East Antarctic Ice Sheet at ~2500 m elevation, down through the Transantarctic Mountains, to the Ross Ice Shelf near sea level. Basemap generated from the MODIS MOA (Scambos et al., 2007) and Antarctic Digital Database via the Quantarctica compilation (http://quantarctica.npolar.no).

To further constrain the pre-Pleistocene configurations of the EAIS, we exploit the extensive moraine record at Roberts
Massif, a high-elevation site in the central TAM, where studies on nearby nunataks have suggested that old (> 5 Ma)
deposits exist (e.g., Ackert and Kurz, 2004). Roberts Massif (86.374°S, 177.135°W) is a ~100 km$^2$ ice-free area
situated at the head of Shackleton Glacier, an outlet of the EAIS (Figure 1). The massif is bounded to the south and
east by the EAIS, to the north and west by the upper Shackleton Glacier, and to the northeast by an unnamed branch
of Zaneveld Glacier. Today, the EAIS at Roberts Massif is cold based and the environment is that of a polar desert.
We employed cosmogenic $^3$He, $^{21}$Ne, $^{10}$Be, and $^{26}$Al to date moraines at Roberts Massif to create a comprehensive
glacial-geologic record for this site comprising 180 samples. Our record affords an unprecedented view of EAIS
variability in the central TAM over the last ~15 Ma and provides valuable new insight into EAIS behavior during
periods of the Miocene and Pliocene, when temperatures and atmospheric $CO_2$ were likely similar to or higher than
today.



## 2 Methods

### 2.1 Geomorphic Mapping and Sample Collection

Fieldwork took place during the 2015–2016 and 2016–2017 austral summers. In the field, we identified and mapped moraines, till deposits, and fault scarps on to 2 m-resolution satellite imagery provided by the Polar Geospatial Center, University of Minnesota. We collected samples for surface-exposure dating from the upper surfaces of erratic boulders located on moraine crests and drift sheets, focusing on boulders in stable positions (i.e., perched atop other boulders, not broken) and exhibiting minimal evidence for surficial erosion. Owing to the prevalence of nuclide inheritance documented by previous Antarctic cosmogenic studies (e.g., Stone et al., 2003; Todd et al., 2010), which is linked to incomplete erosion by cold-based ice of previously exposed surfaces, we sampled large (generally > 1 m tall), angular boulders, following the reasoning that such forms are (i) less likely to have been reworked from the underlying Sirius Group tills than visibly molded, striated, and/or polished cobbles of exotic lithologies, and (ii) more likely to have at least one side that is free of inherited nuclides.

We collected samples of ~1–5 cm thickness using either a hammer and chisel or drill and wedges. To characterize each sampled boulder fully and document its geomorphic context, we described, measured, sketched, and photographed each boulder from at least four different angles. We located samples in the field using an uncorrected handheld GPS unit (estimated horizontal precision typically ± 6 m), and measured elevations by barometric traverse from temporary benchmarks established using differentially corrected GPS and corrected to orthometric heights relative to the EGM96 geoid. The estimated vertical precision of the temporary benchmarks is between ± 0.05 and ± 0.3 m. For barometric differential elevation measurements relative to the benchmarks, we used a Kestrel 4000 barometric altimeter and looped between samples and benchmarks to correct for time-dependent changes in atmospheric pressure. The estimated total uncertainty in sample elevations measured using this procedure is ± 2.5 m, reflecting the precision of the DGPS surveys and the barometer, and the reproducibility of differential barometric elevation measurements of representative sites also surveyed by differential GPS in this and other studies. We measured topographic shielding at sample sites using handheld compass and inclinometer and the procedure described by Balco et al. (2008, with accompanying online material).

### 2.2 Cosmogenic-nuclide measurements

#### 2.2.1 Cosmogenic helium-3 analyses

We measured cosmogenic $^3$He concentrations in pyroxene separated from samples of Ferrar dolerite. To separate pyroxenes at the University of Maine Cosmogenic Isotope Laboratory, we followed a modified version of the method described by Bromley et al. (2014). We sieved crushed samples to isolate the 125–250 μm grain size fraction, which was boiled for two hours in 10% $HNO_3$ to remove Fe oxides and other weathering products. We then removed lighter minerals (mostly plagioclase) using a water-based heavy liquid with density 2.94 g/cm$^3$, and leached remaining material in 5% HF to dissolve adhering plagioclase and remove outer surfaces of pyroxene grains potentially enriched in implanted $^4$He from U and Th decay (Blard and Farley, 2008; Bromley et al., 2014). Finally, etched pyroxenes were





passed through a magnetic separator and hand-picked to remove remaining contaminants under a binocular
microscope.
We then measured $^3$He concentrations in clean pyroxene separates at the Berkeley Geochronology Center using the
BGC "Ohio" system, which consists of a MAP 215-50 sector field mass spectrometer with updated detectors and
counting electronics, coupled to a fully automated gas extraction and purification system. Gas extraction on this system
uses a laser "microfurnace" in which ~15-40 mg aliquots of pyroxene, encapsulated in Ta packets, are heated under
vacuum using a 150W, 810 nm diode laser coupled to a coaxial optical pyrometer in a feedback loop allowing control
of the pyrometer temperature. The pyrometer is calibrated by heating a thermocouple in an identical apparatus.
However, note that precise temperature measurement is not necessary for this work. In most cases (Table S2), we
extracted helium in an initial 15-minute heating step at 1225°C, followed by a second 15-minute heating step at
1325°C to ensure complete extraction. The second heating step typically contained 1–5% of total He released. We
added additional heating steps for a few representative samples to test for complete extraction, and found He signals
indistinguishable from blank. Gases released into the extraction line were purified by reaction with SAES getters and
frozen to activated charcoal at 12 K, after which helium was released into the mass spectrometer at 33 K. In all cases,
we measured $^4$He signals on a Faraday cup and $^3$He on a continuous dynode electron multiplier operated in pulse-
counting mode.

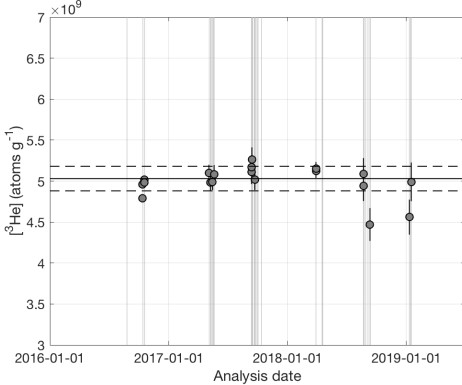
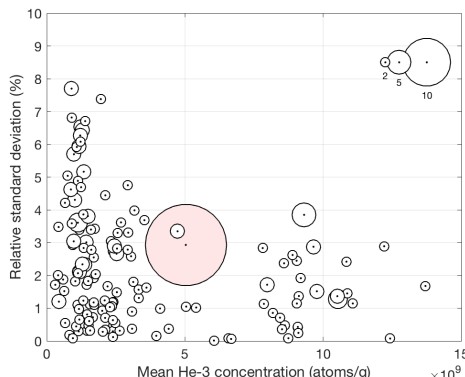

**Figure 2.** Quality-control data for $^3$He measurements. Left panel, replicate analyses of CRONUS-P in all measurement periods during 2016-19. Light gray lines indicate dates samples in this study were analyzed. Error bars show 68% confidence estimates (e.g., 1 sigma); relatively large uncertainties and poor reproducibility in final two measurement periods reflect unusually nonlinear helium sensitivity and relatively large scatter in analyses of gas standards during these periods. Horizontal lines show mean and standard deviation of all measurements. Right panel, relative standard deviation of replicate analyses of 142 samples of Ferrar pyroxene analyzed during this study. 21 of these samples are not from Roberts Massif and therefore are not reported in this study but are included here for completeness. The size of the symbol indicates the number of times each sample was analyzed. The pink circle is CRONUS-P.

We quantified both $^3$He and $^4$He sensitivity by peak height comparison between samples and aliquots of custom-mixed
helium gas standards, calibrated using direct pressure measurements of both isotopes using Baratron capacitance
manometers, containing between 1.57 x 10$^{-18}$ and 4.71 x 10$^{-16}$ moles of $^3$He and between 4.39 x 10$^{-14}$ and 1.26 x 10$^{-11}$



moles of $^4$He. Ferrar pyroxene has relatively high and highly variable $^4$He concentrations, and the MAP 215 mass
spectrometer displays a significant pressure dependence on He sensitivity (Burnard and Farley, 2000), so accurately
quantifying machine sensitivity over a wide pressure range was an important aspect of this work. We addressed this
by (i) source tuning at He pressures similar to those expected for sample analyses to improve linearity in the pressure
range of interest, and (ii) ensuring that observed He pressures in sample analyses were bracketed within the pressure
range available from standard analyses. In many cases, this required discarding results of an initial analysis and
reanalyzing the sample with a different size aliquot calculated to match sample and standard pressures. Total process
blanks measured on empty Ta packets had less than $10^5$ atoms $^3$He and $10^{10}$ atoms $^4$He, which is negligible for all
samples discussed here. Reported measurement uncertainties in $^3$He concentrations include uncertainties from $^3$He
counting statistics (typically 1–2%) as well as the variance in sensitivity inferred from gas standard analyses spanning
the pressure range of interest (typically 1–3%).
As additional quality control measures, we analyzed aliquots of the CRONUS-P pyroxene standard (Blard et al., 2015)
together with samples throughout each period of analysis, and made replicate analyses of a total of 121 pyroxene
samples as well as an additional 21 samples of Ferrar pyroxene from other Antarctic sites (Figure 2). In each of 6
distinct measurement periods between 2016-2019, we analyzed 2-4 aliquots of CRONUS-P. Although average
measured $^3$He concentrations in individual measurement periods varied from $4.80 \pm 0.30$ x $10^9$ atoms/g to $5.14 \pm 0.1$
x $10^9$ atoms/g, data from different measurement periods were not distinguishable as separate populations. The mean
and standard deviation of 19 measurements during the entire period was $5.03 \pm 0.15$ x $10^9$ atoms/g (2.9%), which is
indistinguishable from the accepted value of $5.02$ x $10^9$ (Blard et al., 2015). Replicate analyses of other samples had
a mean relative standard deviation of 2.2% (Figure 2). As expected from counting statistics, replicate scatter varied
with $^3$He concentrations, ranging from 3% for concentrations < 2 x $10^9$ atoms/g to 1.5% for concentrations > 7 x $10^9$
atoms/g.
Ferrar pyroxene is known to contain a non-zero concentration of non-cosmogenic (presumably magmatic) $^3$He. Kaplan
et al. (2017), Margerison et al. (2005), and Ackert (2000) obtained maximum limiting concentrations for non-
cosmogenic $^3$He of 5-–7 x $10^6$ atoms/g, which are consistent with an unpublished estimate (Balco, unpublished data)
of $3.3 \pm 1.0$ x $10^6$ atoms/g. As this is 1.2% of the lowest total $^3$He concentration measured in a Roberts Massif erratic
in this study, and 0.1 % of the average concentration observed, we disregard it and assume that all observed $^3$He in
pyroxene is cosmogenic.

### 2.2.2 Cosmogenic beryllium-10 and aluminum-26 analyses

We purified quartz from sandstone samples using established physical and chemical procedures (e.g., Schaefer et al.,
2009) at the University of Maine Cosmogenic Isotope Laboratory. Chemical extraction of beryllium and aluminum
and preparation of BeO and $Al_2O_3$ targets took place at the University of Maine and Lawrence Livermore National
Laboratory (LLNL). Ratios of $^{10}$Be/$^9$Be were measured relative to the 07KNSTD standard (Nishiizumi et al., 2007) at
LLNL and corrected for background $^{10}$Be by procedural blanks with a range of 23,000–44,000 atoms. Al isotope ratios
are measured relative to the KNSTD standardization of (Nishiizumi, 2004), and corrected for a procedural blank of





75,000 ± 75,000 atoms. Note that blank corrections for both [10]Be and [26]Al are negligible for samples in this study.
One measurement of the CRONUS-A quartz standard (Jull et al., 2015) run together with these samples yielded 3.491
± 0.047 x 10[7] atoms/g [10]Be and 1.494 ± 0.030 x 10[8] atoms/g [26]Al (Table S5), indistinguishable from accepted values
for both nuclides. Reported uncertainties for [10]Be and [26]Al measurements include uncertainties in AMS isotope ratio
measurement, process blanks, and [9]Be/[27]Al concentrations.

### 166     2.2.3 Cosmogenic neon-21 analyses

We measured [21]Ne in the same quartz separates used for [10]Be analysis using the BGC "Ohio" noble gas mass
spectrometer system also used for [3]He measurements and described above. Aliquots of quartz samples were degassed
in two heating steps at 850° and 1100°C, and calculations of excess [21]Ne (see below) are based on total Ne released
in both heating steps. Ne isotope measurements at BGC use a [39]Ar spike to quantify and correct for the [40]Ar[++]
interference on mass 20, and are described in Balco and Shuster (2009). We quantified Ne abundances by peak height
comparison between samples and aliquots of an air standard containing between 5 x 10[-16] and 2 x 10[-14] mol Ne and
calibrated using a Baratron capacitance manometer. In contrast to helium, neon sensitivity was linear within this range
at all times. Corrections for mass discrimination, when necessary, are also based on the air standard and assumed
atmospheric [21]Ne/[20]Ne and [22]Ne/[20]Ne ratios of 0.002959 and 0.1020, respectively. A total of 20 analyses of the
CRONUS-A quartz standard during the period of this study yielded mean and standard deviation of 319.8 ± 6.3
Matoms/g (2% RSD) excess [21]Ne, indistinguishable from the accepted value of 320 Matoms/g (Vermeesch et al.,

178     2015).

Neon isotope ratios, as observed in previous studies for TAM sandstones, were indistinguishable from the
atmospheric-cosmogenic mixing line (see supplementary Table S3). However, Balco et al. (2019) and Middleton et
al. (2012) have also shown that significant concentrations of nucleogenic [21]Ne produced by decay of trace U and Th
are present in quartz from this lithology. To calculate cosmogenic [21]Ne concentrations in quartz samples, therefore,
we first calculated excess [21]Ne with respect to atmospheric composition, followed Balco et al. (2019) in assuming that
excess [21]Ne consists of both cosmogenic and nucleogenic [21]Ne, and estimated nucleogenic [21]Ne concentrations using
the following procedure. First, we measured excess [21]Ne concentrations in a set of six sandstone samples from ice-
proximal sites at upper Roberts Massif that have apparent [10]Be exposure ages less than 10 ka, and one additional
sample with an apparent [10]Be exposure age of 75 ka. Assuming that these samples have experienced a single period
of exposure, we calculated the [21]Ne concentration attributable to this exposure and subtracted it from total excess [21]Ne
concentrations to obtain estimates of nucleogenic [21]Ne; resulting mean and standard deviation for nucleogenic [21]Ne
estimates in these samples are 10.5 ± 2.8 Matoms/g, similar to but slightly higher than estimates for Beacon Group
sandstones in the Dry Valleys region (Balco et al., 2019; Middleton et al. 2012). We then measured U and Th
concentrations in quartz and computed apparent (U-Th)/[21]Ne closure ages as described in Balco et al. (2019);
excluding one outlier attributed to a spurious Th measurement, the mean and standard deviation of apparent closure
ages is 603 ± 110 Ma. If we assume that all other sandstone erratics from Roberts Massif that we analyzed in this
study have a similar source and therefore a similar apparent closure age, we can estimate nucleogenic [21]Ne
concentrations using U and Th concentrations and this closure age estimate. Note that this apparent closure age is





older than the depositional age of the Beacon Group. If these sandstone samples are derived from the Beacon group,
therefore, it is most likely inaccurate as a cooling age. However, the provenance of the sandstone erratics is unknown,
and in any case this inaccuracy would not affect the assumption that Roberts Massif sandstone erratics have a single
characteristic apparent closure age. Table S4 shows the results of this procedure. For samples with less than 200
Matoms/g total excess $^{21}$Ne, we measured U and Th concentrations in individual samples and applied the mean closure
age inferred from the ice-proximal samples, which resulted in subtraction of up to 20% of total excess $^{21}$Ne as
nucleogenic and had a significant effect on results. For samples with higher $^{21}$Ne concentrations, the uncertainty in
the nucleogenic $^{21}$Ne estimate is negligible and we used an average value rather than measuring U and Th in individual
samples. For example, for samples from the Southwest Col on Misery Platform, discussed below, estimated
nucleogenic $^{21}$Ne is less than 0.5% of total excess $^{21}$Ne. Reported uncertainties for $^{21}$Ne measurements, as for $^{3}$He, are
derived from counting statistics as well as reproducibility of the gas standards.

**2.2.4 Treatment of replicates for cosmogenic noble gas measurements**

For the majority of samples, we made replicate $^{3}$He and $^{21}$Ne measurements and performed chi-squared tests on
replicate sets with the null hypothesis that all measurements on the same sample belong to a single population and
disagree only because of measurement uncertainty. If we could not reject the null hypothesis at 95% confidence, we
took the error-weighted mean of replicate analyses as the true nuclide concentration and the standard error as the
uncertainty. If the null hypothesis was rejected, we used the arithmetic mean and standard deviation. A caveat to this
procedure, however, is that we found that our $^{3}$He results from CRONUS-P during the period of this study did not
pass a chi-squared test (p = 0.02), indicating that our internal uncertainty estimates for individual $^{3}$He measurements
are underestimating the true scatter in multiple measurements of the same sample. Thus, we adjusted calculated
uncertainties upward when necessary such that no $^{3}$He concentration has a relative uncertainty less than 2.9%, the
relative standard deviation of CRONUS-P measurements. $^{21}$Ne results from CRONUS-A, on the other hand, passed
the chi-squared test (p = 0.35), so we did not make a similar adjustment to $^{21}$Ne data. However, cosmogenic $^{21}$Ne
concentrations do include an additional uncertainty derived from nucleogenic $^{21}$Ne subtraction after averaging of
replicates.

**2.3 Surface exposure age calculations**

We calculated exposure ages from measured nuclide concentrations using Version 3 of the online exposure age
calculator described by Balco et al. (2008) and subsequently updated (http://hess.ess.washington.edu). We employed
the time-dependent "LSDn" scaling method of Lifton et al. (2014) and the Antarctic atmosphere model of Stone
(2000). Production rate calibration for $^{10}$Be, $^{26}$Al, and $^{3}$He use the "primary" calibration data sets of Borchers et al.
(2016) for these nuclides, and we compute $^{21}$Ne production rates by assuming a $^{21}$Ne/$^{10}$Be production ratio of 4.03
(Balco et al., 2019; Balco and Shuster, 2009b; Kober et al., 2011). In contrast to exposure-dating studies that are
located at similar altitude and latitude to production rate calibration sites, our study involves significant extrapolations
from the locations of calibration data, mostly at low elevation and high latitude or high elevation and low latitude, to
the high-elevation-high-latitude sites at Roberts Massif. Scaling methods that can be fit equivalently to the calibration

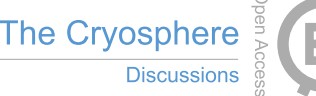

data predict different production rates at our sites. Specifically, production rates predicted by LSDn scaling are ~15%
higher than those predicted by the scaling method of Lal (1991) and Stone (2000) (the 'St' and 'Lm' scaling methods
of Balco et al., 2008). However, at several high-elevation sites in Antarctica, including Roberts Massif, measured [10]Be
and [26]Al concentrations are significantly higher than values for production-decay saturation predicted by the St and
Lm methods, indicating that these methods overpredict production rates at high-elevation-high-latitude locations (see
discussion in https://cosmognosis.wordpress.com/2016/09/09/ saturated-surfaces-in-antarctica/). On the other hand,
saturation concentrations predicted by the LSDn method are consistent with the highest measured [10]Be and [26]Al
concentrations in Antarctica. Thus, we conclude that, at least in the high TAM, exposure ages calculated using LSDn
scaling are likely accurate, and exposure ages calculated using St/Lm scaling would be spuriously old.

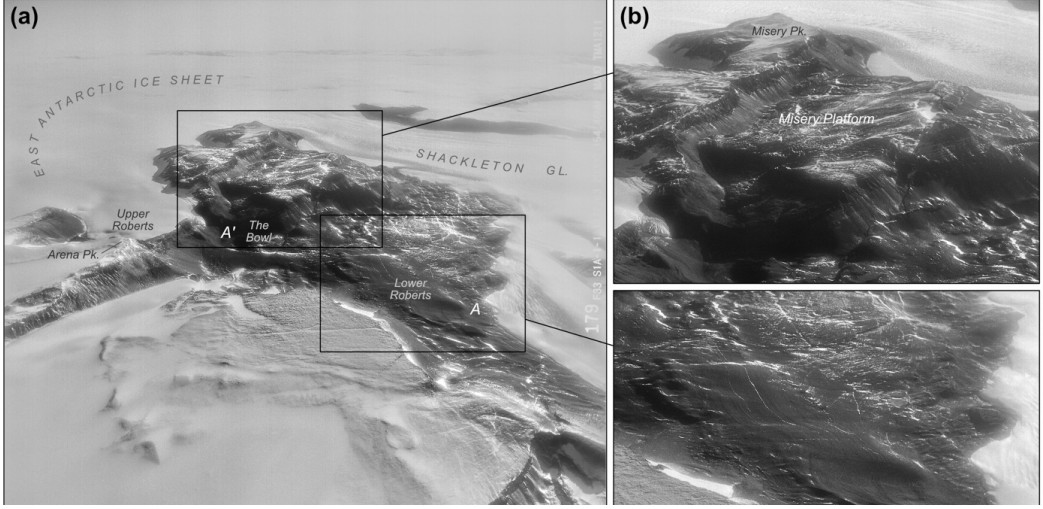

**Figure 3.** Oblique aerial photograph of Roberts Massif looking west along the spine of the Transantarctic Mountains, with the East Antarctic Ice Sheet to the left. Enlargement (A) shows Misery Platform, which is the hanging wall of the large normal fault that bisects the massif. (B) shows the extensive moraine sequence at lower Roberts Massif. The moraine sequence at Upper Roberts (Fig. 7) faces west and is hidden from this viewing angle. A and A' match Figure 4. Image is 1963 U.S. Navy trimetrogon aerial photograph, TMA 1211/179 R.

Additional uncertainties in exposure-age estimates derive from the choice of production rate calibration data.
Estimated total uncertainties for [10]Be exposure ages derived from calibration data are ~6% (Borchers et al., 2016).
Yet, any [10]Be calibration dataset that predicted significantly lower production rates, and therefore lower saturation
concentrations, would not be consistent with the [10]Be data from the Southwest Col (see discussion in section 4.3).
These data permit that we have underestimated [10]Be production rates, but not that we have overestimated them.
However, the majority of data in this study are [3]He exposure ages, and we have no similar constraint on [3]He production
rates. [3]He production rate calibration data display substantially more scatter than [10]Be, and estimates on total global
uncertainty for [3]He exposure dating range from less than 2% (Goehring et al., 2018) to more than 10% (Borchers et
al., 2016; Phillips et al., 2016). Production rate calibration uncertainty therefore may be significant for [3]He results.



## 3 Results

### 3.1 Field Observations



Roberts Massif is defined topographically by large-scale normal faulting that has produced escarpments as much as
~1200 m in relief (Figure 3). These faults delineate a number of broad, sub-horizontal surfaces, including a lower-
elevation platform (hereafter 'Lower Roberts'), a middle-elevation platform, comprising the Misery Platform and
Upper Roberts sites, and the high peaks of the massif, including Misery Peak (2725 m) and Arena Peak (informal
name; 2700 m). Local bedrock comprises sandstones of the Beacon Supergroup and pyroxene-bearing Ferrar dolerite,
which includes a fine-grained variety and a friable, coarse-grained variety. Notably, the termini of the EAIS,
Shackleton Glacier, and the unnamed spur of Zaneveld Glacier at Roberts Massif are relatively free of debris,
containing only the occasional boulder. Further, we did not observe any evidence of glacial outwash or liquid water
at any of these margins, indicating that the ice bounding Roberts Massif is currently cold-based.

### 3.1.1 Lower Roberts


In the southern portion of the Lower Roberts area, a complex of faults forms a deep, back-tilted basin named "The
Bowl" by Hambrey et al. (2003). With the exception of a 100 m-relief bedrock hill, referred to here as the Central
Rise, and the Bowl, the Lower Roberts area exhibits relatively gentle topography (Figure 4). Dolerite bedrock surfaces
outcrop at several locations throughout Roberts Massif, and commonly exhibit glacial polish, striations, and molding
consistent with erosion beneath a wet-based glacier. Most of these bedrock outcrops are directly overlain by semi-
lithified, poorly sorted pockets of sediment (several meters thick in places), containing deeply striated gravel- to
cobble-sized clasts of heterogenous, non-native lithologies embedded in an olive-gray, clay-rich matrix (Figures 4 and
5). We interpret these sediments as lodgement tills associated with the Sirius Group. First described by Mercer (1972),
the Sirius Group occurs throughout the upper (> ~2000 m elevation) TAM as erosional remnants of clay-rich diamicton
that are correlated with at least one period of past temperate glaciation. An in-depth sedimentological study of glacially
eroded bedrock surfaces and Sirius Group tills at Roberts Massif, and other locations along upper Shackleton Glacier,
is provided by Hambrey et al. (2003).
Bedrock and Sirius Group tills are blanketed by patchy glacial drift, comprising primarily angular, cobble-to-boulder-
sized clasts with little-to-no fine-grained material (Figure 5). Ferrar dolerite is the most abundant lithology, although
this drift includes the occasional sandstone boulder, as well as rounded cobbles reworked from the underlying tills
described above. A key feature of this drift deposit is the abundance of open-work boulder moraines, which we
targeted for surface-exposure dating (Figure 6). These low relief (1–2 m high) ridges are composed primarily of large,
angular dolerite boulders and are oriented sub-parallel to the modern ice edge, marking former marginal positions of
the EAIS to the South and the unnamed spur of the Zaneveld glacier to the north. The sediments of these drifts and
associated boulder-belt moraines exhibit characteristics typical of cold-based glaciation, being thin, patchy, and clast-
supported with little-to-no fine grained material (Figures 5 and 6) (Atkins, 2013). Furthermore, clasts are generally
angular and lack the striations, polish, and molding associated with erosive wet-based ice.


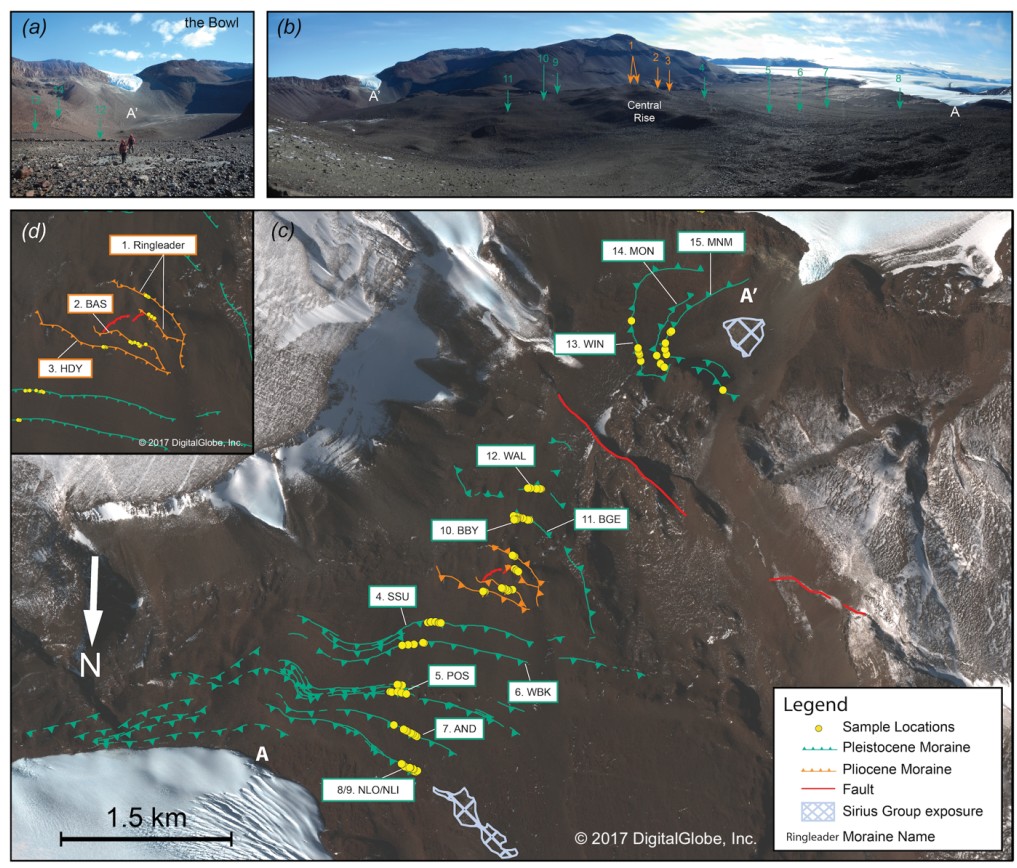

**Figure 4:** Map of Lower Roberts. a) Photograph of the Bowl, showing cold-based drift and moraines overlying Sirius Group deposits, which appear light gray, and b) photo of the Lower Roberts area shown with arrows pointed to sampled moraines, with numbers corresponding to moraine names in (c) and letters A and A' corresponding to positions in (c). c) Glacial geomorphic map showing moraines and sample locations at Lower Roberts, as well as the location of observed Sirius Group outcrops. The basemap is derived from Worldview-2 satellite imagery (copyright 2017, DigitalGlobe, Inc.).

We identified and sampled for surface-exposure dating 15 moraines throughout the Lower Roberts area. We focused
on the most prominent, laterally continuous moraines, which comprise accumulations of stacked boulders, and avoided
the numerous discontinuous moraine mounds and isolated erratic boulders, from which former ice marginal positions
are difficult to reconstruct. The stratigraphically oldest moraine in the Lower Roberts sequence, the Ringleader
moraine (informal name) encircles the summit of the Central Rise, indicating that north- and south-flowing ice masses
once converged to form a continuous ice surface across the Lower Roberts area at least ~170 m higher than the modern
ice margin to the north. From the Ringleader moraine, at the highest position in the Lower Roberts site, we sampled
northern (extending from Ringleader to A in Figure 4) and southern (extending from Ringleader to A' in Figure 4)
moraine transects. Listed in stratigraphic order, the northern transect included the BAS, HDY, SSU, WBK, POS,





AND, and NLO/NLI moraines (moraine initials correspond to informal names and sample ID suffixes listed in the
ICE-D Antarctica online database and Table S1); from the southern transect we sampled the BBY, BGE, WAL, WIN,
MON, and MNM moraines. Notably, the POS moraines constitute a complex of three main ridges, while the NLO/NLI
moraines comprise two distinct ridges spaced only by ~5 m.
The youngest deposit at Roberts Massif comprises a thin layer of sandstone and dolerite debris that extends several
tens of meters beyond the current ice margins. Clasts are relatively unweathered (i.e., exhibit minimal staining and/or
exfoliation), and exhibit fresh scuff marks [abrasions formed as cold-based ice drags entrained boulders across
underlying surfaces (Atkins et al., 2002)] (Figure 5f). With the exception of a few discontinuous segments, this unit
generally is not associated with distinct moraines. Based on strong similarities in position, morphology, and relative
weathering with deposits reported from other TAM sites (e.g., Todd et al., 2010), we correlate the youngest drift unit
at Roberts Massif with the most recent Late Quaternary expansion of Shackleton Glacier/EAIS and do not discuss it
further.
Outboard of this relatively unweathered limit, drift and moraine boulders become progressively more weathered with
distance from and elevation above the modern ice. For instance, dolerite boulders belonging to the outermost deposits
of the HDY, BAS, and Ringleader moraines (up to 3 km from and 170 m above the modern ice margin) exhibit dark
red staining, pitting of up to ~0.5 cm depth, exfoliation up to ~4 mm, and weathering rinds 1–2 mm thick, while the
presence of sandstone clasts is increasingly rare (Figure 6d). In contrast, dolerite boulders that we sampled on the
innermost moraines were generally blue-grey in color and lacked significant weathering characteristics, such as
staining or pitting (Figure 6c). Although the boulders on the outermost moraines at Roberts Massif display more
pronounced weathering than those on the inner moraines, the characteristics described here represent relatively
minimal surface weathering compared to slightly warmer and wetter Antarctic locations, such as the McMurdo Dry
Valleys. There, ~3 Ma clasts, which are similar in age to those on the HDY, BAS, and Ringleader moraines (Section
3.4), display pitting greater than 4 cm depth (Swanger et al., 2011). Additionally, we did not observe any cross-cutting
relationships between moraine crests throughout Lower Roberts, either on the ground or in satellite imagery.
Therefore, we conclude that moraines at this site increase in age with distance away from and elevation above the
modern ice sheet surface. Altogether, these surface-most deposits indicate that the Lower Roberts area records > 15
prior expansions of cold-based ice.



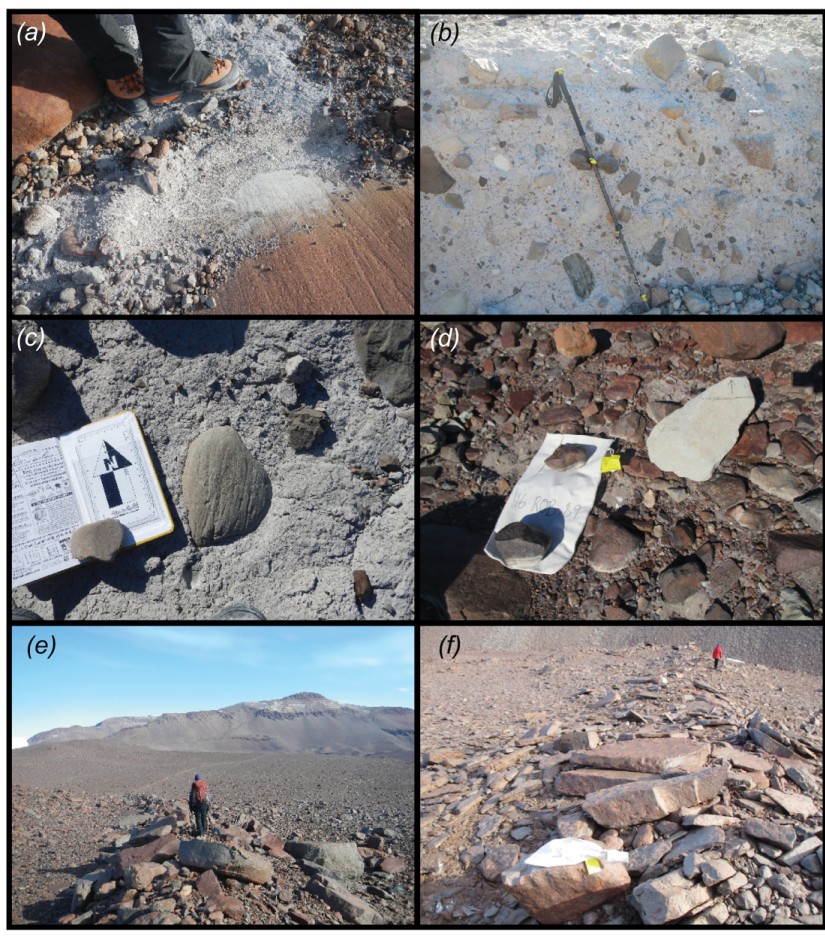

**Figure 5:** Views of drifts and tills described at Roberts Massif. a) The gray, fine-grained Sirius Group deposits atop striated dolerite bedrock; b) Sirius Group exposed in section in the Bowl; c) Striated, glacially molded Sirius cobble embedded in a fine-grained matrix; d) Sample 16-ROB-089-COL, a freshly-scoured sandstone clast in the Bowl, likely deposited as a thin drift sheet atop older deposits during a Late Quaternary expansion of the EAIS; e) cold-based AND moraine, which is Pleistocene in age; and f) Misery B moraine, which is Miocene in age.

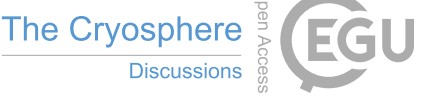

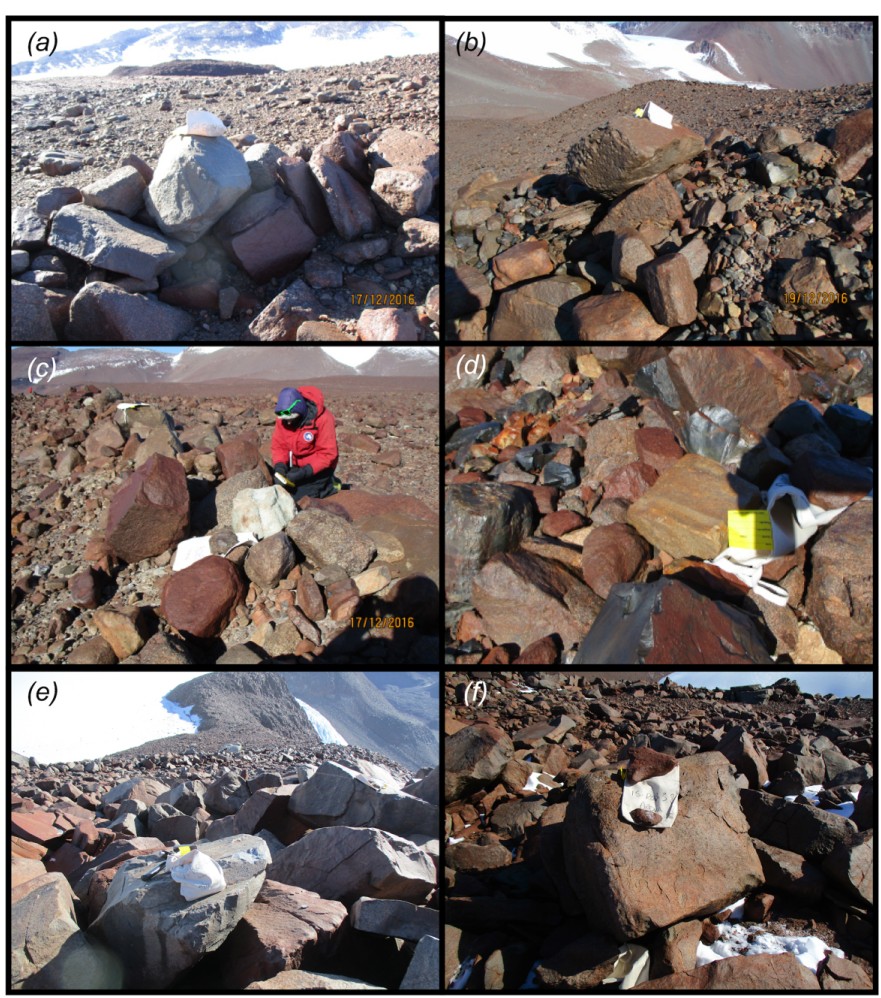

**Figure 6:** Photographs of moraines and sampled boulders at Roberts Massif. a) Blue-gray dolerite boulder 16-ROB-010-NLO on the second moraine from the modern EAIS in the Lower Roberts northern transect; b) Red-stained dolerite boulder 16-ROB-059-RIN on the Ringleader moraine, the outermost moraine in the Lower Roberts area; c) Relatively unweathered sandstone boulder 16-ROB-009-NLO; d) Red-stained/varnished sandstone boulder 16-ROB-062-RIN on the Ringleader moraine; e) Relatively unweathered, blue-gray dolerite boulder 15-ROB-064-MUS on the Musik moraine, the innermost moraine at Upper Roberts; and f) Weathered/red-stained dolerite boulder 15-ROB-038-ARM on the Arena moraine, the outermost moraine at Upper Roberts.



### 3.1.2 Upper Roberts

The Upper Roberts site is situated on a steep, west-facing slope of Arena Peak, directly adjacent to the northward flowing lobe of the EAIS that ultimately flows over the Bowl headwall (Figure 7). Here, we mapped glacial drift and moraines identical in character to those at Lower Roberts, indicating deposition by a cold-based EAIS. Similar to observations at Lower Roberts, a fresh-looking drift of sandstone and dolerite boulders extends several tens of meters beyond the modern ice edge. At the Upper Roberts site, that fresh deposit is associated with a low-relief (~1.5 m) ridge. We attribute this deposit to the most recent expansion of the EAIS during the Late Quaternary and do not discuss it further in this paper. We focused on five moraine ridges located along a vertical transect between ~60 m and 150 m above and oriented sub-parallel to the modern ice surface (2150 m). In order of descending elevation, we identified and sampled the Arena (2300 m), Eine (2260 m), Kleine (2240 m), Nacht (2220 m), and Musik (2220 m) moraines (informal names). Additionally, we mapped moraine segments preserved both within and above (up to ~2500 m elevation) this transect, but, owing to lateral discontinuity and poor preservation on high-gradient slopes, we did not sample these limits for surface-exposure dating. As at Lower Roberts, the general increase in boulder-surface weathering and the absence of cross-cutting moraine stratigraphy (determined from field observations and satellite imagery) suggests that glacial deposits at Upper Roberts become older with increasing elevation above the modern EAIS (Figure 6).

### 3.1.3 Misery Platform

Misery Platform is a broad, gently sloping platform in the southwest part of Roberts Massif (Figures 3 and 8). Comprising the top surface of the hanging-wall block of a large normal fault, Misery Platform is bounded to the south by a ~300–340 m-high fault scarp. At the base of the scarp, we mapped a series of arcuate moraine ridges (here termed the Misery moraines), four of which we sampled for exposure-age dating (Figures 8, 9). The southern edge of the footwall block, which includes Misery Peak (2723 m elevation), drops steeply to the EAIS surface at ~2200 m elevation, and exhibits south-facing, amphitheater-shaped valleys that are occupied partially by north-flowing lobes of the EAIS (Figure 8). The largest of these valleys is located directly south of the Misery Moraines, and its extension above the current surface of the EAIS suggests that this lobe of ice was significantly thicker in the past. Further, a thin drift of glacial erratics atop the footwall block at ~2550 m elevation mark where a north-flowing lobe of the EAIS overtopped the broad slopes east of Misery Peak and cascaded down the escarpment, where it deposited the Misery moraines on the platform below. This interpretation requires that the Misery moraines (a) postdate the formation of the fault scarp and (b) were deposited by an EAIS that was sufficiently thick (> 300 m above the current surface) to overtop the footwall block. Although the Misery moraines are similar in elevation to those sampled at Upper Roberts, they represent the highest former ice surface elevation of the EAIS examined in this study.

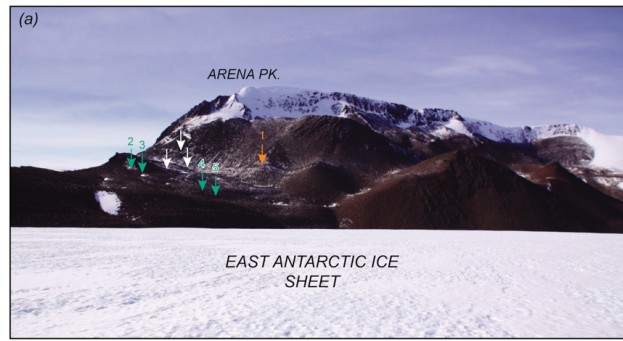

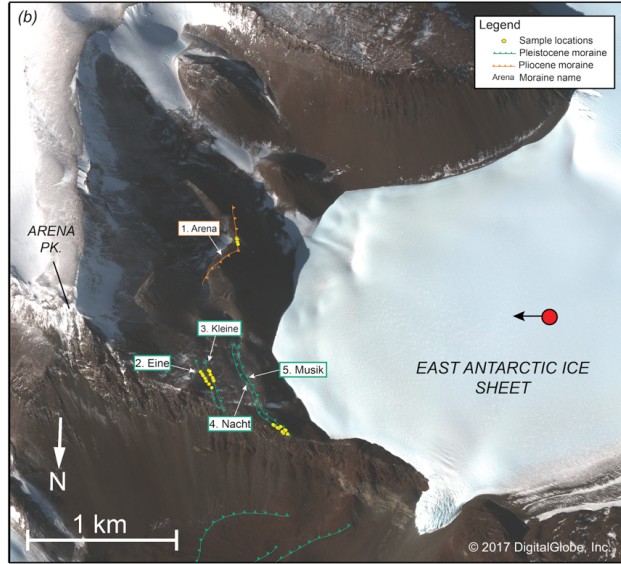

**Figure 7.** Upper Roberts Massif. a) Photograph of the Upper Roberts transect with moraines marked by arrows, numbered corresponding to sampled moraines in (b). White arrows in (a) denote undated moraines. b) Geomorphic map of Upper Roberts. The red circle and arrow shows the location and vantage of photo in (a). The basemap in (b) is derived from Worldview-2 satellite imagery (copyright 2017, DigitalGlobe, Inc.).

Compared to moraines at Lower and Upper Roberts, the Misery moraines are relatively broad and high-relief (~2–5
m high) and comprise finer matrix material (silt-to-gravel). Moraine crests are mantled with angular dolerite boulders
exhibiting pronounced weathering features, including deep red-to-purple staining, 2–3 mm-thick weathering rinds,
and ventifaction pits of up to 2 cm depth. On the basis of these physical characteristics, they appear older than the
outermost moraines at both Lower and Upper Roberts. Therefore, we interpret the Misery moraines as cold-based ice-
marginal features marking the ostensibly oldest and most extensive EAIS terminus positions that we documented at
Roberts Massif. We used cross-cutting relationships of the Misery moraines to determine their stratigraphic order.
From outermost (oldest) to innermost (youngest), we sampled boulders on the following moraine crests: Misery D,
Misery A, Misery B, Misery C (note that the designations A-D are field designations reflecting the sequence of sample
collection, not the stratigraphic order; Figure 8). Importantly, we avoided sampling adjacent to overlapping moraine
segments.





Immediately outside of, and stratigraphically underlying, the Misery moraines, the weathered bedrock surface is
mantled with a thin patchy ablation till, dominated by dolerite boulders and a small number of sandstone clasts, and
associated with a coarse-grained sand and gravel deflation surface. We observed this unit throughout Misery Platform
and collected samples for surface-exposure dating from boulders on Southwest Col, located approximately 1.5 km
northwest of the Misery moraine complex and 400 m above the modern surface of Shackleton Glacier (Figure 8).
Here, the ablation till ('Southwest Col drift') mantles a bedrock surface of heavily stained and deeply exfoliated
coarse-grained dolerite. In places, granular sediments fill joints and depressions in the bedrock. These sediments are
characterized by red-stained silt-to-gravel-sized grains, which may derive from the disintegration of the dolerite
bedrock, and gravel-to-cobble-sized clasts of various lithologies. In contrast to the Sirius Group deposits observed
elsewhere at Roberts Massif, boulders comprising Southwest Col drift are predominantly dolerite (as opposed to a
broad mix) and generally more angular.
We sampled three dolerite clasts (1 boulder and 2 cobbles) and four sandstone clasts (3 boulders and 1 cobble), all of
which are perched on bedrock and/or interstitial sediments, for surface-exposure dating. The surface of the dolerite
boulder (15-ROB-28-COL) exhibits deep red staining and evidence of significant wind abrasion, except on the lee
side where there is a thick red-brown weathering rind (Figure 9). The sandstone boulders (15-ROB-32-COL, 15-ROB-
33-COL, and 15-ROB-34-COL) exhibit orange-to-red staining, surface varnish, and ventifaction of up to 4 cm depth.
Based on the thin nature of this deposit, we interpret the Southwest Col drift as a cold-based ablation till deposited by
the EAIS. Owing to its weathering state, we suggest that this deposit is the oldest glacial unit in our record. Surface-
exposure ages from this site therefore provide a minimum-limiting age for temperate glaciation at Roberts Massif.
**3.1.4 Summary of Field Observations**
We mapped three primary surfaces at Roberts Massif (listed in stratigraphic order): glacially molded and striated
dolerite bedrock, temperate-style tills belonging to the Sirius Group, and cold-based drifts associated with openwork
boulder moraines. All samples collected for surface-exposure dating are derived from the cold-based deposits marking
former positions of the EAIS. At both the Lower and Upper Roberts sites, weathering patterns and the lack of cross-
cutting moraines suggest that relative moraine ages increase with distance from, and elevation above, the modern ice
sheet margin. Deposits on Misery Platform (the Misery moraines and the Southwest Col drift) exhibit more advanced
subaerial weathering than our other sites, indicating that these deposits are significantly older. In Section 4.2, we
describe results from cosmogenic-nuclide measurements made on samples from 23 separate moraine ridges and one
drift sheet.



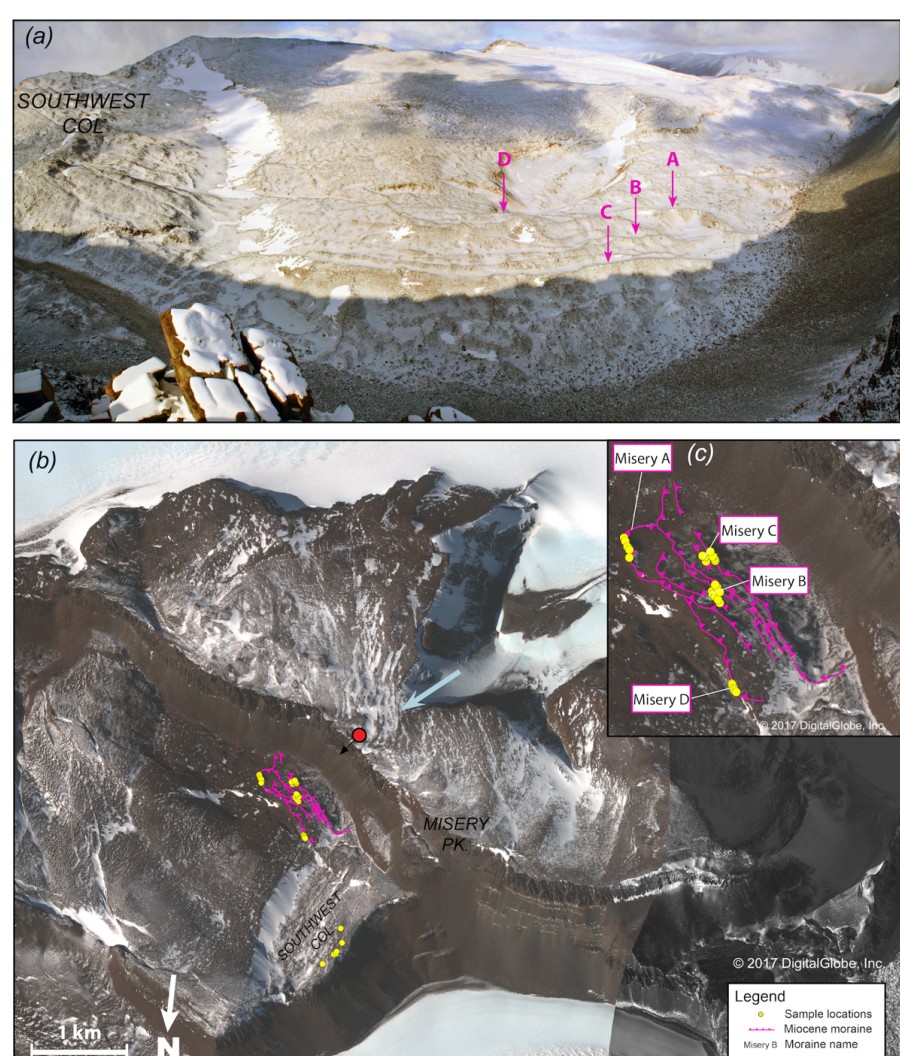

**Figure 8:** Map of Misery Platform. a) Photo of the Misery Moraines. Pink arrows point to the sampled Misery Moraines and are labelled with the corresponding moraine letter. The location of the Southwest Col drift is also labeled. The photo was taken from the location of the red circle in (b) looking in the direction of the black arrow (vantage to the northeast). b) Geomorphic map of the Southwest Col area. The Southwest Col Drift mantles the bedrock outboard of the Misery moraines. The blue arrow denotes the direction of ice flow when the Misery Moraines were deposited. The basemap in (b) and (c) is derived from Worldview-2 satellite imagery (copyright 2017, DigitalGlobe, Inc.).

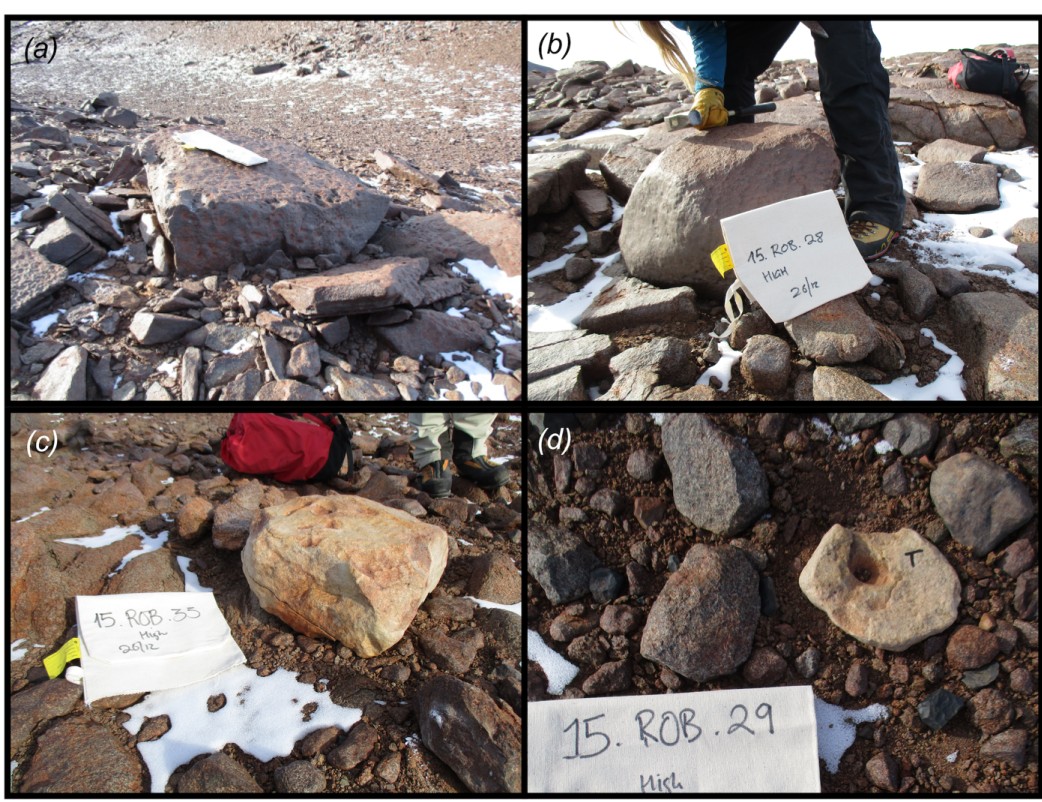

**Figure 9:** Photographs of boulders from the Misery Platform. a) Dolerite boulder 15-ROB-017-MZC on the Misery C moraine; b) Dolerite boulder 15-ROB-028-COL at the Southwest Col; c) Sandstone boulder 15-ROB-035-COL at the Southwest Col; d) Sandstone cobble 15-ROB-029-COL at the Southwest Col.

### 3.2 Results from Cosmogenic-Nuclide Measurements

We made cosmogenic [3]He measurements in pyroxene from 167 dolerite boulders; 38 [21]Ne and 13 [10]Be measurements in quartz from 13 sandstone boulders; and two [26]Al measurements in quartz from two sandstone boulders (also measured for [21]Ne and [10]Be). Samples were derived from 23 distinct moraine crests and one glacial drift sheet (Southwest Col). Apparent exposure ages span two periods: ~13–8 Ma at Misery Platform and ~3 Ma–400 ka at Upper and Lower Roberts (Tables 1 and S1). "Apparent" exposure ages refer to the calculated age of the boulder given the measured nuclide inventory, assuming that the boulder has experienced only one period of exposure, with no erosion or burial during that time. Boulder information, nuclide concentrations, complete step-degassing results for [3]He and [21]Ne are summarized in Tables S2, S3, and S5, and the full dataset is archived online in the ICE-D:ANTARCTICA database (http://antarctica.ice-d.org). In this section, we summarize these cosmogenic-nuclide data and highlight the





possible effects of surface erosion and other geomorphic processes on exposure ages, which ultimately lead us to
estimates of the emplacement age of the moraines.

### 3.2.2 Constraints on erosion rates from paired $^{10}$Be-$^{21}$Ne measurements

As the majority of landforms at Roberts Massif are several million years old, quantifying the magnitude of surface
erosion is key to accurate exposure-dating. Here, we summarize geochemical data and field observations that allow
us to place limits on long-term erosion rates. Four sandstone erratics at Southwest Col have $^{10}$Be concentrations close
to predicted production-erosion saturation values, and apparent $^{21}$Ne exposure ages of 9–12 Ma. As these samples
have nearly the highest concentrations of these nuclides yet measured on Earth, concentration measurements are
correspondingly (and unusually) precise, making it possible to use the paired $^{10}$Be/$^{21}$Ne data to simultaneously infer
exposure ages and surface erosion rates from these samples (Figure 10) (Gillespie and Bierman, 1995; Lal, 1991).
Given the assumption that these samples have experienced continuous exposure at a steady erosion rate, the $^{10}$Be/$^{21}$Ne
data imply true exposure ages in the range 12–15 Ma, but varying surface erosion rates in the range 0.5–3 cm/Myr.
These low erosion rates are consistent with our field observations pertaining to surface erosion of these sandstones as
described in Section 3.1.3.
Apparent $^{3}$He exposure ages from three dolerite clasts also located on Southwest Col, and which therefore should have
the same true exposure age as the sandstone clasts, are 8.6 Ma, 10 Ma, and 11 Ma. Assuming that the true exposure
age of the deposit is no greater than 14.5 Ma, as implied by the two-nuclide data for the highest-nuclide-concentration
sandstone (15-ROB-032-COL) shown in Figure 10, this implies maximum erosion rates for the dolerite clasts of 3.8,
2.7, and 1.9 cm/Myr, respectively. Further assuming that the dolerite clast with the highest $^{3}$He concentration (15-
ROB-028-COL) has been exposed at the drift surface for the longest period, and has therefore experienced mainly
surface weathering rather than exhumation from till, we propose that ~2 cm/Myr is likely a maximum limit on rock
surface erosion rates for dolerite surfaces in our study area. The assumption that this clast has been exposed at the
surface is supported by the fact that 15-ROB-028-COL is a boulder, while the rest of the dolerite surfaces we sampled
on Southwest Col are cobbles. If the deposit is younger than 14.5 Ma, an even lower erosion rate would be implied.
Although this is an extremely low surface weathering rate by global standards, it is nonetheless consistent with the
polar desert climate and the field observations described in section 3.1.3 (i.e., angular clasts with surface varnish and
minimal pitting).

### 3.2.3 Information about geomorphic processes from multiple-nuclide measurements

As on Southwest Col, we also measured multiple nuclides ($^{10}$Be and $^{21}$Ne, and, in one case, $^{26}$Al) in several sandstone
boulders on the Ringleader, WIN, MON, AND, and NLO moraines at Lower Roberts (Figure 11). Although sandstone
clasts are rare on these moraines, these data provide some insight into the exposure history of these boulders that we
can use to assess the importance of inheritance and post-depositional disturbance for moraine exposure ages.



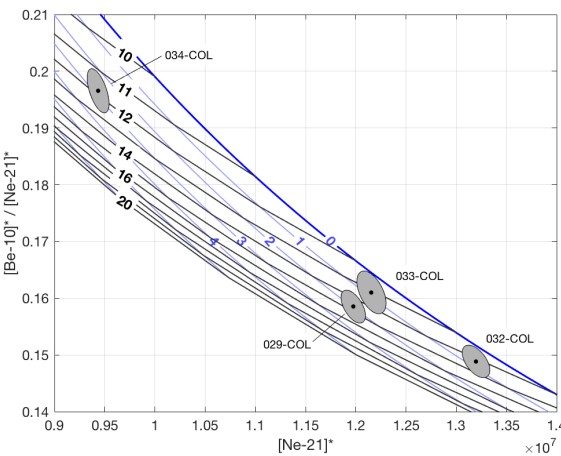

**Figure 10.** $^{10}$Be-$^{21}$Ne normalized two-nuclide diagram for Southwest Col sandstone erratics. Blue lines are isolines of constant steady erosion (cm/Myr); black lines are isolines of constant exposure age (Ma). The diagram is constructed using LSDn production rate scaling and a $^{21}$Ne/$^{10}$Be production ratio of 4.03 (Balco et al., 2019). Note that the x-coordinate, the $^{21}$Ne concentration normalized to the production rate, is equivalent to the apparent $^{21}$Ne exposure age. Although apparent $^{21}$Ne exposure ages for these samples are 9.5–13 Ma, the two-nuclide diagram shows that the data are better explained by 12–15 Ma exposure at erosion rates between 0.5–3 cm/Myr.

In general, a boulder that has experienced a single period of exposure that is equal to the emplacement age of the
moraine should display concordant $^{10}$Be, $^{21}$Ne, and $^{26}$Al ages that are the same as those of other boulders on the
moraine. For the Ringleader moraine (Figure 11), $^{10}$Be-$^{21}$Ne-$^{26}$Al measurements are concordant at 2.8–3 Ma, therefore
consistent with simple exposure at negligible erosion, and lie in the center of the range of $^{3}$He ages from dolerite clasts
on the same moraine (Figure 12). These observations suggest that (i) the sandstone boulders have experienced a single
period of exposure with minimal post-depositional exhumation or weathering, which is consistent with our field
observations as described in section 3.1.1, (ii) their exposure age most likely represents the true emplacement age of
the moraine, and (iii) two outliers in the $^{3}$He age distribution can likely be attributed to both inheritance (one ~4 Ma
age) and post-depositional disturbance (one ~2 Ma age) .
In contrast, paired $^{10}$Be-$^{21}$Ne measurements on four boulders on the MON moraine and one on the WIN moraine
(Figure 11), both adjacent to the Bowl and emplaced by ice from upper Roberts overflowing the Bowl headwall (Figure
4), display discordant apparent ages. Additionally, apparent exposure ages from both sandstone and dolerite boulders
at these moraines are relatively scattered (coefficient of variance > 20%). The $^{10}$Be-$^{21}$Ne data (Figure 11) could be
explained either (i) by an extended period of steady erosion at an ice-free site prior to entrainment and deposition of
the clasts, or (ii) by repeated exposure and ice cover of the samples prior to emplacement. Both of these conditions
are likely if these boulders were sourced from the adjacent outcrop area of sandstone on the Bowl headwall (Figure
4a). Thus, we consider it most plausible that the apparent exposure ages of these sandstones reflect prior exposure
and, thus, overestimate the true age of the moraine. In general, these results imply that high scatter in exposure ages



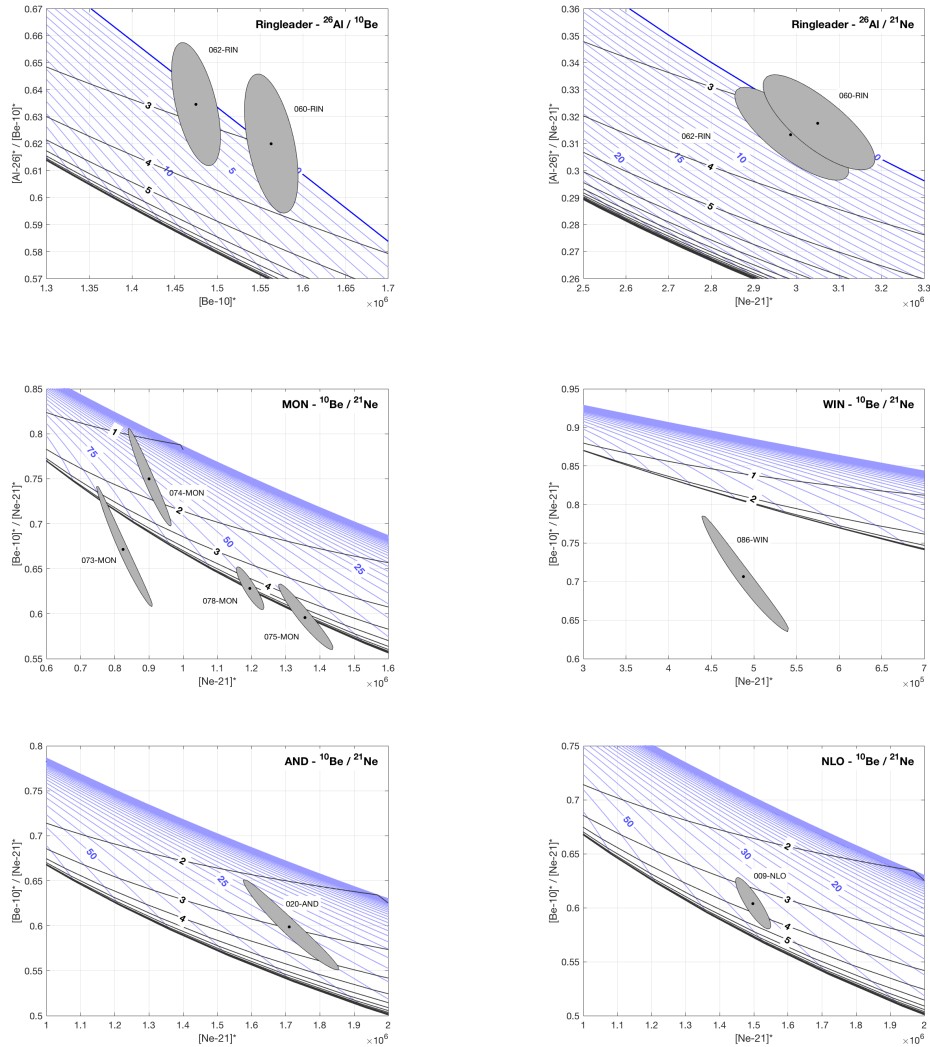

**Figure 11.** Two-nuclide diagrams for all sandstone erratics collected from lower Roberts Massif moraines. The construction of the diagrams is the same as in Fig. 10. Two-nuclide data for sandstones on the Ringleader moraine lie on the simple exposure line and are in agreement with ³He ages, suggesting that these samples experienced a single period of exposure at negligible erosion, and their apparent ages are a good estimate of the true age of the moraine. On the other hand, paired nuclide data from sandstones on the MON, WIN, AND, and NLO moraines require either significant erosion or a multistage exposure history. An erosion explanation would predict that their apparent ages should be younger than ³He ages on the same moraines; as this is not the case, these samples most likely experienced a multistage exposure history and therefore were emplaced with significant nuclide inheritance.

for moraines in the Bowl are most likely explained by inherited nuclide concentrations in clasts sourced from the
adjacent headwall, and the true ages of the moraines are therefore likely close to the young end of their age
distributions.



Finally, paired $^{10}$Be-$^{21}$Ne measurements from the AND and NLO moraines (Figure 11), both at the ice-proximal end
of the northern Lower Roberts transect, fall within the "erosion island" on the two-nuclide diagram, indicating that
their true exposure ages are older than the apparent ages for either nuclide. In addition, these clasts have apparent ages
higher than most $^{3}$He ages from these moraines (Figure 12). Again, this is best explained if the scatter exhibited by
these moraines is largely the result of inheritance.
Overall, although we have a relatively small number of multiple-nuclide data from sandstone boulders, our results
demonstrate that (i) inheritance is unequivocally present in some moraine boulders and (ii) inheritance is likely most
significant at moraines where boulders are likely sourced from a combination of far-traveled EAIS subglacial debris
and cliff fall within the massif itself. These scenarios are also consistent with the observation that boulders on moraines
at upper Roberts, which can only be derived from beneath the EAIS, exhibit substantially less scatter than moraines
at lower Roberts (Table 1 and Figure 12), where additional input from rockfall is likely. Overall, while none of our
observations exclude post-depositional disturbance as a potential source of scatter, they do show that inheritance is
likely a more important contributor.
**3.3 Outlier Elimination**
For each moraine dated, we measured cosmogenic nuclides in 6–8 individual clasts. We observed a variety of
distributions ranging from tightly grouped age sets, which likely reflect dispersion due to measurement uncertainties
alone, to highly scattered distributions with both old (indicative of nuclide inheritance) and young outliers (e.g., due
to subaerial weathering and/or post-depositional disturbance, such as rock toppling or cracking). To interpret these
age distributions and arrive at realistic estimates of the moraine age, we utilized constraints from field observations,
the stratigraphic ordering of the moraines, exposure-age trends across moraine transects, and measurements of
multiple nuclides in various clasts (see above).
We first considered geomorphic stratigraphy, weathering characteristics, and trends in exposure-age distributions to
identify and eliminate outliers. For Misery Platform, we utilized the cross-cutting relationships of the Misery moraines,
which elucidate relative age, to identify exposure ages that are outliers. Although we did not observe such cross-
cutting relationships at Upper and Lower Roberts, we exploited the fact that both apparent exposure ages and physical
weathering state increase with distance from and elevation above the modern ice margins to determine relative ages
of the moraines, and thus to identify likely outliers.
We performed an initial screening to remove outliers by assuming that the true depositional age of each moraine lies
within the range of measured exposure ages on this moraine. If true, then any exposure ages on one moraine that are
older than all exposure ages on a stratigraphically older moraine must be erroneous. Likewise, any exposure ages that
are younger than all ages on a stratigraphically younger moraine must also be erroneous. Applying this rule recursively
to stratigraphically ordered sets of moraines resulted in the rejection of 46 measurements on 22 boulders (Figures 12
and 13; Table S1). We also rejected 9 measurements on 5 boulders as outliers likely resulting from geomorphic
processes (i.e., inheritance or post-depositional disturbance), which were not rejected as stratigraphic outliers yet are
> 2σ beyond the main age population on that moraine (see Table S1). After this stratigraphic screening was complete,



we also rejected as outliers 14 non-concordant ---[10]Be and [21]Ne measurements on 10 sandstone boulders located on
the NOLO, AND, WIN, and MON moraines, as those boulders likely contain inherited nuclides (see discussion in
Section 3.2.3). In total, we rejected 69 measurements on 37 boulders (Figures 12 and 13; Table S1).
The resulting boulder age distributions for each moraine exhibit a variety of forms. Many moraines (e.g., Arena, BAS,
Misery B moraines; Figures 12 and 13) display a central cluster approximating a normal distribution, and for these
moraines we assign the mean and standard deviation of the ages as the best estimate of the depositional age of the
moraine. Other moraines (e.g., SSU, WAL, BGE) showed heavily skewed, bimodal, or scattered age distributions; for
these we provide age ranges rather than means in the discussion that follows. In the case of those high-scatter moraines,
it is likely that the true moraine age is closer to the younger end of the age range, as we identified inheritance as a
more likely contributor to moraine scatter than post-depositional disturbance (Section 3.2.3).
**3.4 Moraine ages**
In this section, we summarize moraine age estimates assuming zero surface erosion (Table 1; Figures 12 and 13); we
discuss the effects of this assumption in later sections.
**Lower Roberts:** The oldest dated moraine in the Lower Roberts area – Ringleader –dates to 2.94 ± 0.24 Ma. Along
a northward transect from the summit of the Central Rise to the modern ice margin, subsequent moraines yielded the
following ages (moraine initials correspond to informal names and sample ID suffixes listed in the ICE-D Antarctica
online database; Figure 11): BAS (2.94 ± 0.14 Ma), HDY (2.84 ± 0.08 Ma), WBK (1.62–2.84 Ma), SSU (1.90–2.95
Ma), POS (1.16–2.05 Ma), AND (1.08–1.63 Ma), NLO (1.07–1.58 Ma), NLI (0.54–2.09 Ma). A similar transect
extending southward from the Central Rise provides the following moraine ages: BBY (1.55–2.69 Ma), BGE (1.41–
2.93 Ma), WAL (1.50–2.80 Ma), WIN (0.51–1.00 Ma), MON (0.54 ± 0.01 Ma), and MNM (0.40–0.87 Ma). As
discussed in Section 4.2.4, this southern transect displays the highest degree of age scatter, potentially due to the
incorporation of rockfall from the surrounding escarpments.
**Upper Roberts:** Moraine ages at Upper Roberts display a high degree of internal consistency and are reported here
from highest moraine to lowest: Arena (2.64 ± 0.13 Ma); Eine (1.19 ± 0.14 Ma); Kleine (1.18 ± 0.16 Ma); Nacht (1.11
± 0.10); and Musik (0.61–1.10 Ma) (Figure 11). As noted in Section 4.1, undated moraine segments located above the
Arena moraine represent higher surface levels of the EAIS, potentially prior to ~2.6 Ma. Additionally, undated
moraine segments situated between the Arena and Eine moraines, which differ in elevation by ~45 m, may account
for the temporal gap between these two limits.
**Misery Moraines:** Approximately 1.5 km southeast of the Southwest Col drift (~14.5 Ma, section 3.2.2), the Misery
moraines yielded ages (listed from outermost moraine to innermost) of 7.94 ± 0.23 Ma (Misery D; n = 4), 7.93 ± 0.23
Ma (Misery A; n = 1), 7.99 ± 0.06 Ma (Misery B; n = 8), and 7.63 ± 0.29 Ma (Misery C; n = 5) (Figure 12). We
consider a young population of ages, between ~4 and 6 Ma, on the Misery A and Misery C moraines to be outliers as
the bulk of ages from the complex cluster around 8 Ma. Due to the excellent internal consistency of these age
populations, we consider it unlikely that the 8 Ma population reflects inheritance, as that mechanism typically
introduces considerable scatter to the data set (Balco, 2011).





**Table 1.** Roberts Massif moraine and drift ages and statistics.

| Site | Elevation (m) | Count (samples exluded) | Age Range of Raw Data (Ma) | Mean Age (Ma)[1] | Age Range (Ma)[2] | Coefficient of Variance (%) | Reduced $\chi^2$ |
|---|---|---|---|---|---|---|---|
| ***Misery Platform*** | | | | | | | |
| Southwest Col | 2377 | 7 (4) | 5.20 – 12.86[3] | - | 8.63 – 12.86 | - | - |
| Misery D | 2249 | 4 (1) | 7.43 – 8.21 | 7.94 ± 0.23 | - | 3% | 1.00 |
| Misery A | 2198 | 1 (4) | 4.34 – 7.93 | 7.93 ± 0.23 | - | '- | '- |
| Misery B | 2252 | 8 (0) | 7.88 – 8.08 | 7.99 ± 0.06 | - | 1% | 0.07 |
| Misery C | 2215 | 7 (2) | 4.70 – 7.96 | 7.63 ± 0.29 | - | 4% | 1.74 |
| ***Upper Roberts*** | | | | | | | |
| Arena | 2303 | 6 (0) | 2.50 – 2.85 | 2.64 ± 0.13 | - | 5% | 2.49 |
| Eine | 2255 | 5 (2) | 0.89 – 2.07 | 1.19 ± 0.14 | - | 11% | 14.23 |
| Kleine | 2241 | 6 (0) | 0.97 – 1.37 | 1.18 ± 0.16 | - | 14% | 23.74 |
| Nacht | 2221 | 6 (1) | 1.03 – 1.52 | 1.11 ± 0.10 | - | 9% | 6.42 |
| Musik | 2215 | 3 (0) | 0.61 – 1.10 | - | 0.61 – 1.10 | 30% | 184.80 |
| ***Lower Roberts*** | | | | | | | |
| Ringleader | 1957 | 9 (2) | 2.16 – 4.07 | 2.94 ± 0.24 | 2.57 – 3.37 | 8% | 5.10 |
| *Northern Transect* | | | | | | | |
| BAS | 1914 | 7 (0) | 2.76 – 3.18 | 2.94 ± 0.14 | 2.76 – 3.18 | 5% | 2.52 |
| HDY | 1895 | 5 (2) | 2.09 – 3.48 | 2.84 ± 0.08 | 2.75 – 2.97 | 3% | 0.90 |
| WBK | 1877 | 6 (1) | 1.62 – 3.66 | - | 1.62 – 2.84 | 22% | 81.95 |
| SSU | 1872 | 7 (0) | 1.90 – 2.95 | - | 1.90 – 2.95 | 15% | 28.78 |
| POS | 1865 | 8 (0) | 1.16 – 2.05 | - | 1.16 – 2.05 | 21% | 48.23 |
| AND | 1830 | 6 (3) | 0.89 – 1.66 | - | 1.08 – 1.63 | 15% | 24.88 |
| NLO | 1829 | 6 (2) | 1.07 – 1.58 | - | 1.07 – 1.58 | 19% | 36.07 |
| NLI | 1832 | 4 (3) | 0.54 – 2.09 | - | 0.54 – 1.39 | 36% | 373.32 |
| *Southern Transect* | | | | | | | |
| BBY | 1905 | 5 (0) | 1.55 – 2.69 | - | 1.55 – 2.69 | 25% | 68.30 |
| BGE | 1906 | 6 (1) | 1.41 – 4.12 | - | 1.41 – 2.93 | 27% | 157.67 |
| WAL | 1896 | 7 (0) | 1.50 – 2.80 | - | 1.50 – 2.80 | 23% | 69.22 |
| WIN | 1818 | 7 (2) | 0.38 – 1.00 | - | 0.51 – 1.00 | 20% | 76.27 |
| MON | 1791 | 2 (8) | 0.53 – 1.38 | 0.54 ± 0.01 | - | 2% | 0.24 |
| MNM | 1774 | 4 (3) | 0.40 – 1.85 | - | 0.40 – 0.87 | 33% | 166.30 |

[1] Used for moraines with age distributions close to normal, and/or a coefficient of variance <15%

[2] Used for moraines with non-normal (scattered) age distributions which have a coefficient of variance >15%

[3] Four youngest ages, obtained from $^{10}$Be measurements, are near saturation and thus not considered.

[4] Population statistics only presented for moraines·

[5] All boulders but one are considered outliers on Misery A·



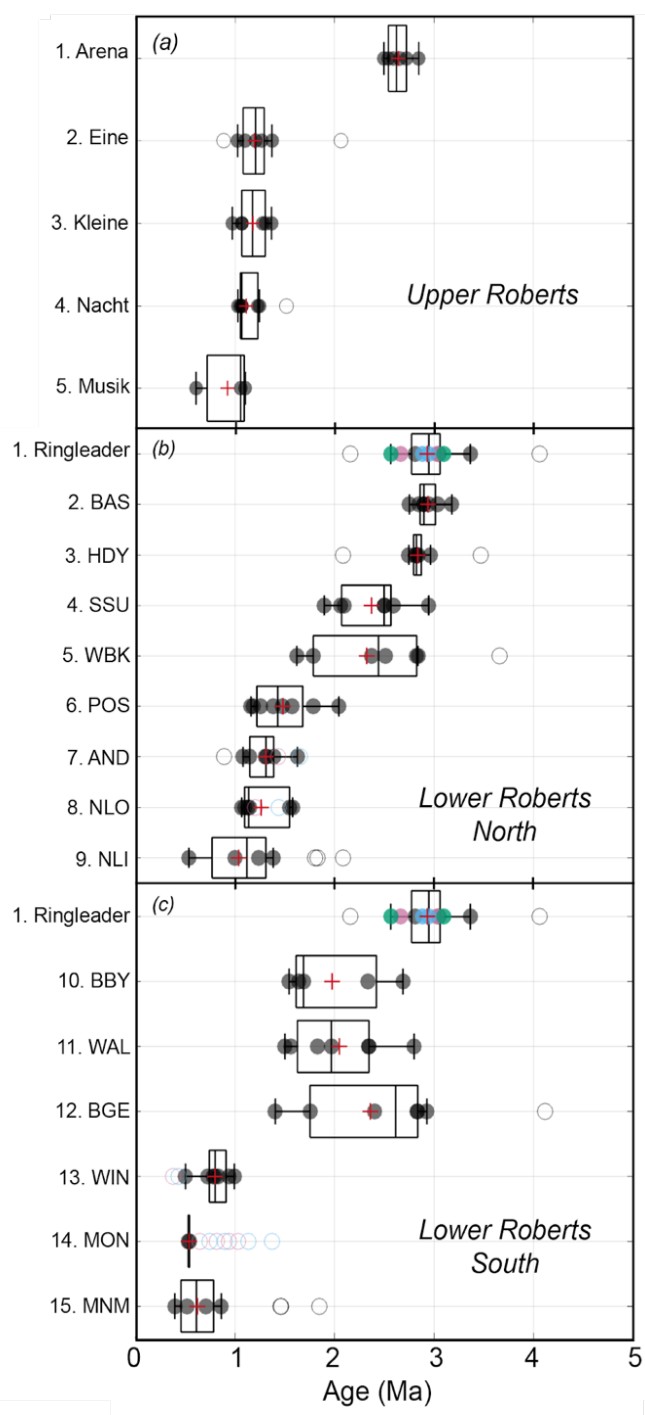

**Figure 12:** Boxplots showing moraine ages for the Plio-Pleistocene part of the Roberts Massif record. Moraines for each site (Upper Roberts and Lower Roberts northern and southern transects) are listed in stratigraphic order, with the outermost moraine at the top of each panel. Moraine numbers in (a) correspond to those in Figure 7, while moraine numbers in (b) and (c) correspond to those in Figure 4. The Ringleader moraine is shown in both panels (b) and (c), as it is the uppermost moraine in both Lower Roberts transects. [3]He ages are black, [21]Ne ages are blue, [10]Be ages are pink, and [26]Al ages are green. Outliers are shown as open circles. The average moraine age is denoted by a red plus symbol.
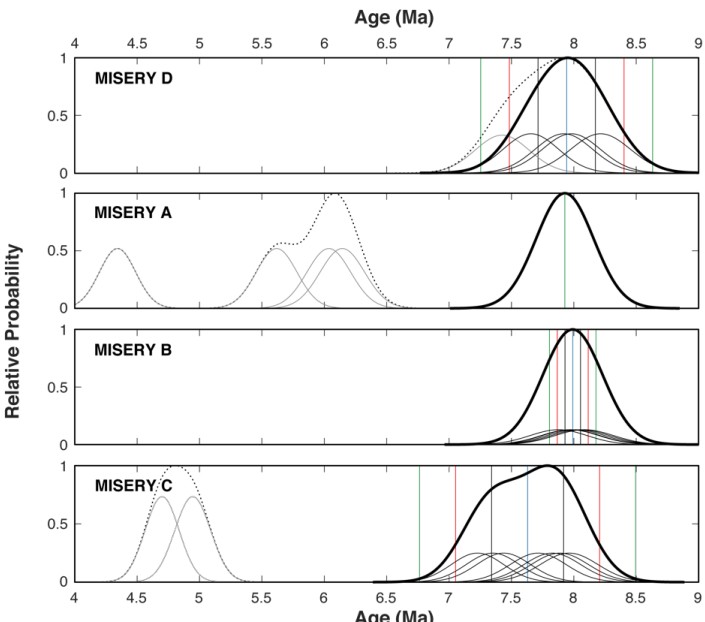

**Figure 13:** Camel plots (i.e., normal kernel density functions) for the four Misery moraines. The arithmetic mean of the reduced dataset is denoted by the blue line, while the 1σ, 2σ, and 3σ uncertainty envelopes are shown in black, red, and green, respectively. Dotted black lines show the summed probability distributions for the full dataset, including outliers shown in gray, while the thick black lines show the probability distribution for the reduced dataset.

## 4 Discussion

Cosmogenic-exposure ages on moraines and glacial drift at Roberts Massif afford unprecedented insight into Late Cenozoic variability of the EAIS. The record begins at ~14.5 Ma (Southwest Col drift), while distinct ice-marginal positions date to ~8 Ma and between ~3–1 Ma. As described in Section 4.1, all moraines are characteristic of cold-based glacial conditions and are oriented sub-parallel to the modern EAIS margins, suggesting ice configuration similar to today and the buttressing presence of grounded ice in the Ross Sea Embayment (e.g., Alonso et al., 1992; Bromley et al., 2010; Hauptvogel and Passchier, 2012). In the following sections, we discuss the length of the Roberts Massif glacial-geologic record and address the climatic implications of our findings.

### 4.1 Uplift at Roberts Massif

Relative to previous glacial-geologic archives from Antarctica, the Roberts Massif record is exceptionally long (~14.5 Ma). At all three sites described in Section 4, moraine age increases with distance from and elevation above the modern



EAIS, with the oldest site (Misery Platform) ostensibly indicating the thickest ice. One hypothesis to explain the age-
elevation relationship at Roberts Massif is that, while the massif itself has remained isostatically stable for duration of
our record, the surface elevation of the EAIS during glacial maxima has lowered systematically over time.
Alternatively, the configuration of the EAIS during glacial maxima has remained roughly constant for the duration of
the record, but the underlying bedrock has undergone uplift due to tectonism, dynamic topography, and/or isostasy,
processes relevant to the millions-of-years timescale. Tectonic uplift at Roberts Massif since ~15 Ma likely was
minimal; apatite fission thermochronology in the central TAM suggests that major faulting due to tectonism was
complete by ~30 Ma (Fitzgerald, 1994; Miller et al., 2010). However, over the last 3 Myr, approximately 40 m of
uplift at Roberts Massif may be attributed to dynamic topography (Austermann et al., 2015), though this value cannot
account fully for the ~3 Ma ice positions situated ~170 m (Ringleader moraines) and ~180 m (Arena moraine) higher
than the modern EAIS at Lower and Upper Roberts, respectively.
Instead, isostatic rebound resulting from deepening of outlet glacier troughs (i.e., removal of rock and replacement by
less dense ice) may account for much of the apparent moraine elevation loss through the Roberts Massif record. While
large portions of EAIS outlet glaciers, including Shackleton Glacier, are likely frozen to the bed, and thus minimally
erosive, regions of these glaciers are thick enough to be at the pressure melting point today (Golledge et al., 2014),
and thus eroding their beds. Removal of several hundred meters of rock since the mid-Miocene would therefore result
in isostatic rebound of a few hundred meters (Wateren et al., 1999). Although we cannot quantify total trough erosion
over the course of our record, this magnitude of uplift is consistent with the observed elevational offset between relict
moraines and the modern EAIS.
As well as elucidating deposition age, near-saturation concentrations of [10]Be on Southwest Col (15-ROB-033-COL)
and [26]Al the Ringleader moraine (16-ROB-062-RIN) afford maximum-limiting values for isostatic uplift at Roberts
Massif, both since ~14.5 Ma and during the last 3 Myr. For these samples, [10]Be and [26]Al concentrations become
saturated (with respect to LSDn scaling) with erosion rates of ~2.3 g cm$^{-2}$ Myr$^{-1}$ and ~7 g cm$^{-2}$ Myr$^{-1}$, respectively. If
we assume that this apparent erosion rate reflects not removal of mass by surface weathering, but rather a decrease in
atmospheric depth due to uplift, these erosion rate values provide maximum uplift rates. The [10]Be saturation erosion
rate for 15-ROB-033-COL yields an uplift rate of ~24 m Myr$^{-1}$ over the last ~14 Myr, indicating that the total
maximum uplift over the course of the record is ~350 m, or ~70 m over the last 3 Myr. This estimate accounts for less
than half of the elevation difference between the ~3 Ma Ringleader moraine and the EAIS margin (~170 m). In
contrast, the [26]Al saturation erosion rate for 16-ROB-062-RIN affords a higher uplift rate of ~70 m Myr$^{-1}$ over the last
3 Myr, or ~210 m over the Plio-Pleistocene portion of the record, a value that accounts for the full ~170 m elevation
difference between the Ringleader moraine and the modern EAIS. Importantly, both the 24 m Myr$^{-1}$ and 70 m Myr$^{-1}$
values each represent maximum uplift rates under the assumption of zero erosion, meaning that the average pace of
uplift during the Plio-Pleistocene may not have differed from that during the last ~14.5 Ma. In fact, because [26]Al does
not quite reach saturation in 3 Myr, it is likely that the 70 m Myr$^{-1}$ is an overestimate. Moreover, the true uplift rate at
Roberts Massif probably was lower than those calculated here, since our field observations indicated that some, albeit
minor, post-depositional surficial erosion has taken place (Section 3.1).




Uplift of ≤ ~200 m over the Plio-Pleistocene is consistent with cosmogenic-nuclide concentrations from the McMurdo
Dry Valleys, which indicate minimal vertical change during this period (Brook et al., 1995). Similarly, $^{40}Ar/^{39}Ar$ ages
on subaerial volcanic cones limit uplift to 300 m in the Dry Valleys over the past 3 Ma (Wilch et al., 1993) and < 67
m in the Royal Society Range over the past 7.8 Ma (Sugden et al., 1999). In contrast, Stern et al. (2005) posit that > 1
km of isostatic uplift throughout the central TAM has occurred since 35 Ma due to glacial erosion. If true, the
cosmogenic-nuclide concentrations presented here imply that nearly all of this uplift must have taken place between
35 and 14 Ma.
Given the likelihood of isostatic uplift over the long duration of our record, which potentially accounts for much of
the offset between moraine elevations and the modern EAIS, we cannot evaluate changes in ice thickness throughout
this ~14 Myr record with certainty. However, we emphasize that a large, cold-based ice sheet with configuration
similar to today was present during the dated parts of this record.

### 4.2 Miocene presence of the EAIS

The oldest dated glacial unit at Roberts Massif, Southwest Col drift, was deposited ~14.5 Ma and demonstrates that
the EAIS in the central TAM was cold-based by at least the mid-Miocene (Figure 13). This finding aligns closely with
earlier work from the northern TAM that placed the transition to polar conditions at ~14–15 Ma (Denton and Sugden,
2005). We note that deposition of Southwest Col drift also coincided broadly with a mid-Miocene climatic shift
documented in the Olympus Range, McMurdo Dry Valleys, where well-preserved terrestrial and lacustrine fossils
interbedded with ash fall deposits have been interpreted as reflecting an 8°C cooling of Antarctic summers at ~14.5
Ma (Lewis et al., 2008). In addition, the age of Southwest Col drift, which provides a minimum-limiting age for cold-
based glaciation in the central TAM, is approximately coeval with the Mid-Miocene Cooling Transition (~15–13 Ma),
marked by a decline in global sea-surface and bottom-water temperatures (Lear et al., 2015) and atmospheric $CO_2$
concentrations (Zhang et al., 2013). Finally, Southwest Col drift affords minimum-limiting age constraint for the
underlying Sirius Group till at Roberts Massif and supports previously published surface-exposure data suggesting
that these temperate deposits are > 5 Ma (Ivy-Ochs et al., 1995; Schaefer et al., 1999).
Overlying Southwest Col drift, the ~8 Ma Misery moraines represent the oldest ice-marginal landforms identified at
Roberts Massif and suggest the presence of a large, cold-based ice sheet at that time. This EAIS configuration is
broadly coincident with elevated sea-surface temperatures (Herbert et al., 2016) and Antarctic Bottom Water
temperatures (Lear et al., 2015), and potentially higher atmospheric $CO_2$ (Sosdian et al., 2018) relative to the Plio-
Pleistocene. Therefore, our record suggests that a substantial EAIS occupied the central TAM at ~8 Ma despite
generally warmer-than-present climatic conditions (Figure 14).

### 4.3 Plio-Pleistocene presence of the EAIS

The majority of moraines in the Roberts Massif record date to ~3–1 Ma, thus documenting the persistence of a large
EAIS during the Plio-Pleistocene transition and early Pleistocene (Figure 13). Because the uncertainties in our moraine
ages (~0.1–0.5 Ma) exceed the 40-kyr climate cycles dominant during the pre-MPT world, we do not assign moraines



to individual climate events, such as Marine Isotope Stages (i.e., Lisiecki and Raymo, 2005; Railsback et al., 2015).
Nonetheless, moraines dated to > ~2.5 Ma indicate a large EAIS in the central TAM during times when global
temperatures and atmospheric $CO_2$ were likely higher than today (Willeit et al., 2019).
Several moraines at Roberts Massif date to ~3 Ma (Ringleader, ~3 Ma; BAS, ~3 Ma; HDY, ~2.8 Ma; Arena, ~2.6
Ma), inviting the question of whether any of these landforms correspond to the Mid-Pliocene Warm Period (MPWP:
~3.3–3.0 Ma), which has garnered attention as a plausible analog for modern anthropogenic warming. The ongoing
debate regarding the resilience of the EAIS during the MPWP bears two leading hypotheses: (i) that the EAIS was of
similar extent, or potentially larger, than today during the MPWP (e.g., Sugden et al., 1993; Winnick and Caves, 2015)
due to increased East Antarctic precipitation under warmer atmospheric conditions (Huybrechts, 1993); and (ii) that
the EAIS was significantly smaller than today (Scherer et al., 2016; Webb et al., 1984) as a result of enhanced melting
along marine margins (Pollard and DeConto, 2016) and associated structural collapse (Pollard et al., 2015). At Roberts
Massif, moraines dating to the MPWP would support the first hypothesis; however, an absence of MPWP moraines
neither proves nor disproves the second hypothesis, as geologic evidence for even a slightly smaller EAIS would lie
beneath the modern ice sheet surface (Balco, 2015). Below, we address the possibility that any Roberts Massif
moraines date to the MPWP, given the uncertainties associated with exposure dating (i.e., erosion, production rate
error, and uplift).
First, we address the possibility that erosion of boulder surfaces, which acts to remove a portion of the cosmogenic
nuclide inventory, yielded erroneously young apparent exposure ages for the Late Pliocene moraines. As shown in
Section 3.2.3, concordant [10]Be-[21]Ne-[26]Al measurements on Ringleader sandstones afford an exposure age of ~3 Ma,
consistent with the [3]He ages on that moraine, and both sandstone and dolerite boulders appear to have experienced
relatively minimal erosion (i.e., angular, minimal pitting and exfoliation; Section 3.1.1). Applying the maximum
surface erosion rate for dolerites of 2 cm/Myr, determined using the [3]He concentration of 15-ROB-028-COL (Section
3.2.2), the average dolerite age on Ringleader is 3.18 Ma and thus within the uncertainty of the apparent moraine age.
Together, our field observations and cosmogenic-nuclide measurements suggest that the apparent age of the
Ringleader moraine is not erroneously young due to surface erosion. As discussed in Section 4.1, the maximum
possible error in moraine age due to uplift is the same as that for erosion, meaning that the inclusion of uplift has no
significant impact on moraine age.
Next, we explore the potential effect of cosmogenic nuclide production-rate uncertainty on moraine age. The [10]Be
production rate is accompanied by ~6% error and [3]He by ~10% error (Borchers et al., 2016), meaning that the
Ringleader moraine could be ~6 % older (with a lower production rate) or younger (with a higher production rate),
using the more precise [10]Be production rate as a limit (note: [10]Be and [3]He ages are statistically indistinguishable).
However, we can use the boulder with the highest [10]Be concentration on Southwest Col (15-ROB-033-COL), which
is close to saturation, to provide a lower limit for the [10]Be production rate. Applying a production rate ~2% lower than
the globally calibrated production rate of Borchers et al. (2016), which we used to calculate the [10]Be ages presented
here, sample 15-ROB-033-COL becomes oversaturated with respect to LSDn scaling, suggesting that, at most, the
Ringleader moraine (the oldest in the Plio-Pleistocene sequence) is no older than ~3 Ma. Conversely, if the true





production rate is higher than that of Borchers et al. (2016), it is possible that the Ringleader moraine is up to 6%
younger (~2.8 Ma) than reported here. As there are no sandstones on the oldest landform in the Upper Roberts
sequence – Arena Moraine (~2.7 Ma) – we assess the full 10% range in $^3$He production rate. Assuming a 10% reduction
in production rate, the Arena moraine could date to ~3 Ma, or the end of the MPWP.
In summary, we did not date any moraines unequivocally to the MPWP, suggesting that the EAIS was not significantly
larger than today during that time. However, given the dataset presented here, we cannot evaluate further the
configuration of the EAIS during the MPWP because evidence for the ice sheet extent during that time lies beneath
the modern glacier. Moreover, we note that our moraine chronology lacks landforms dating to the earlier Pliocene (~5
Ma), when conditions are thought to have been as warm as during the MPWP (Burke et al., 2018). Nevertheless, our
current dataset provides evidence for a large, cold-based EAIS in the central TAM during the Late Pliocene,
immediately following the MPWP, and in the early-to-mid Pleistocene.
**5 Conclusions**
Surficial deposits characteristic of cold-based glaciation at Roberts Massif span the Last Glacial Maximum to Mid-
Miocene, thereby providing an exceptionally long geologic record of glaciation for the central TAM. The preservation
of numerous, vertically offset ice-marginal deposits is most plausibly explained by the persistence of an EAIS similar
in configuration to today during multiple glacial maxima, accompanied by gradual isostatic uplift of Roberts Massif.
Coupled with extremely low erosion rates (<< 5 cm/Myr), the prevalence of cold-based deposition over the last ~14.5
Ma supports persistent polar desert climate conditions in East Antarctica since the mid-Miocene. Our record also
provides minimum-limiting age control for the underlying Sirius Group deposits, suggesting that at least some of the
temperate glacial deposits preserved in the TAM are older than 14.5 Ma.
Although the Roberts Massif record is not a direct measure of East Antarctic ice volume, our dataset indicates that the
EAIS was not any larger during the late Pliocene-early Pleistocene than it was during parts of the Miocene, even
though temperatures cooled progressively through the Plio-Pleistocene. Nonetheless, the absence at Roberts Massif
of ice-marginal deposits dating unequivocally to the MPWP highlights a critical area for continued investigation, since
distal paleoclimate evidence and model simulations suggest the EAIS was smaller than present at that time. Accepting
that geologic evidence for even a slightly smaller EAIS during the MPWP would lie beneath the modern ice sheet, we
cannot further evaluate the extent to which the EAIS was smaller during the MPWP with the current data set from
Roberts Massif.
In summary, the Roberts Massif dataset provides a long-term, terrestrial perspective of ice sheet extent in the central
TAM, and shows that the EAIS has been a persistent feature of this region since the mid-Miocene. Throughout this
record, the EAIS  has maintained a configuration similar to today, which requires grounded ice the Ross Sea
Embayment, and by extension, West Antarctica, even during periods when global temperature and atmospheric $CO_2$
concentrations likely were similar to or higher than present.



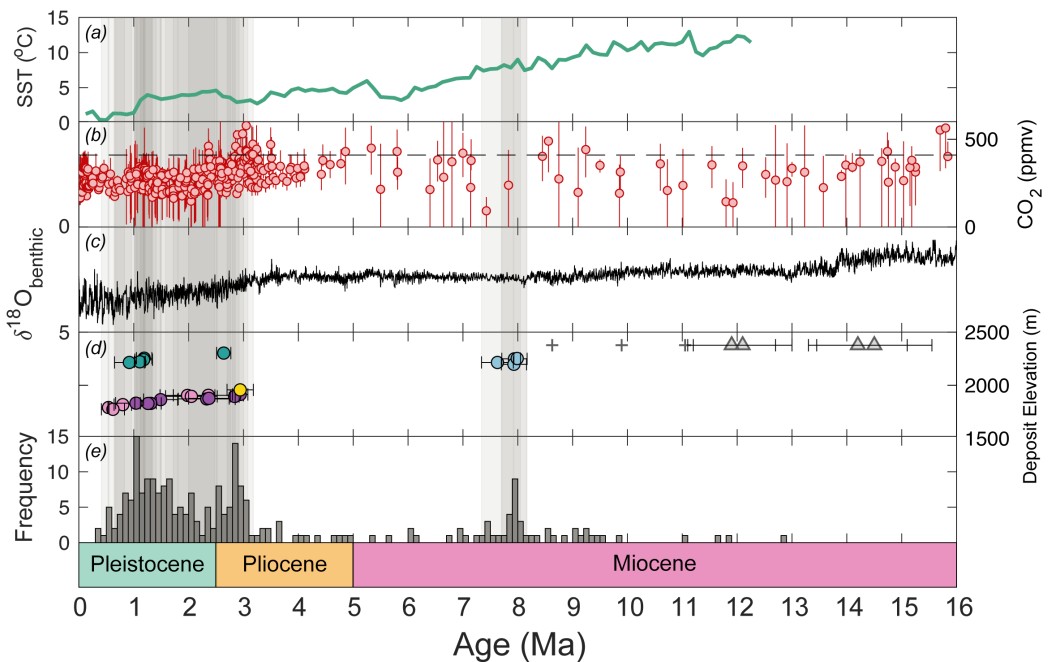

**Figure 14**: Comparison between Roberts Massif glacial chronology and relevant climate records. (a) Southern Hemisphere alkenone-derived temperature stack (Herbert et al., 2016); (b) Boron-isotope-, paleosol-, and stomata-derived $CO_2$ records (Beerling et al., 2009; Breecker and Retallack, 2014; Da et al., 2019; Dyez et al., 2018; Ji et al., 2018; Sosdian et al., 2018; Wang et al., 2015; Zhang et al., 2013) (d) benthic oxygen isotope stack (De Vleeschouwer et al., 2017), (e) Moraine age and uncertainty at Roberts Massif, plotted against deposit elevation. Note that deposition of the Misery moraines required ice to be > 300 m thicker than today, which is not reflected in the moraine elevation. Blue circles are the Misery moraines, teal circles are Upper Roberts moraines, the yellow circle is the Ringleader moraine, purple circles are moraines in the Lower Roberts northern transect, pink circles are moraines in the Lower Roberts southern transect, gray plus signs are apparent ages of dolerite boulders of the Southwest Col Drift, and gray triangles are the age of the Southwest Col sandstones, accounting for erosion, (f) Histogram of all apparent exposure ages at Roberts Massif, including outliers. Colors on the timescale at the bottom correspond to moraine colors in Figures 4, 7, and 8. Vertical gray bars denote moraine ages, including uncertainty. Darker gray color shows a higher frequency of moraines.

**Data Availability**
All analytical information associated with cosmogenic-nuclide measurements appear in the supplementary tables.
Analytical information, with additional sample documentation and photographs, is also available in the ICE-
D:ANTARCTICA online database (http://antarctica.ice-d.org/).



**Author Contribution**

All authors conducted fieldwork, sample collection, and sample preparation for cosmogenic-nuclide analyses. Balco, Balter and Thomas carried out cosmogenic noble gas measurements, and were responsible for data reduction and analysis. Balter prepared the manuscript with contributions from Balco and Bromley.

**Competing Interests**

The authors declare that they have no conflict of interest.

**Acknowledgements**

This work was supported by U.S. National Science Foundation grants ANT-1443329 and ANT-1443321 and by the Ann and Gordon Getty Foundation. It would not have been possible without major contributions from many elements of the U.S. Antarctic Program, including the 109th Airlift Wing of the New York Air National Guard, pilots and ground crews of Kenn Borek Air, and many USAP staff at Shackleton Glacier Camp and McMurdo Station. In addition, we thank Chris Simmons for field mountaineering support, Tim Becker for assistance with noble gas measurements at BGC, Kaj Overturf for help with sample crushing and sieving at the University of Maine, and Brenda Hall for insightful discussions. Geospatial support for this work was provided by the Polar Geospatial Center under NSF-OPP awards 1043681 and 1559691.

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
