# Peer review of "A 14.5 million-year record of East Antarctic Ice Sheet fluctuations from the central Transantarctic Mountains, constrained with cosmogenic 3He, 10Be, 21Ne, and 26Al"

_The Cryosphere, 2020_

## Referee Comment (RC1) · David Sugden (Referee) · 30 Mar 2020

Significance and novelty This paper uses cosmogenic isotope analysis in an imaginative and thorough way to establish that moraines in the central Transantarctic Mountains have been accumulating for at least 15 Ma. This is an important finding. The implication is that the East Antarctic Ice sheet has been present in approximately its present form for the same period. This means it has survived warmer than present climate interludes such as those of the Miocene and Pliocene. The glaciological significance of the finding is twofold. First the moraines tell of the behaviour of the ice sheet close

to its centre, namely the head of Shackleton Glacier, a major outlet glacier traversing the Transantarctic Mountains. Second, the results confirm earlier indications of East Antarctic ice-sheet persistence for 15 Ma based in volcanic ash dating in the different location of the McMurdo Dry Valleys. Two locations using two different forms of dating adds traction to the idea of a persistent East Antarctic ice sheet since the Miocene. The paper is also significant in the way cosmogenic isotope analysis can be used investigate extremely old exposure histories and rates of change. The researchers analysed 180 samples from 24 deposits, using four isotopes. The moraines form an orderly topographic sequence with the highest moraines furthest away from the ice margin being the oldest ($\sim$ 15 Ma) and the lower ones closer to the ice margin with ages of 3.0 - 0.4 Ma. Although no specialist in the intricacies of cosmogenic isotope analysis, I was impressed by the care in sampling and the targeted use of different isotopes to establish confidence limits, to identify and explain outliers, and to yield reliable exposure ages. The results yield the lowest erosion rates on earth at 0.5 – 3.0 cm per million years. Moreover the long time scale was used to test such ideas as the rate of mountain uplift. The record of relative elevation change represented by the decline of ages towards the glacier margin is argued to reflect isostatic uplift of the mountains as a result of erosion of bedrock by glaciers. Differential tectonic uplift is ruled out because the oldest moraines drape over fault scarps. Specific comments 1. Is it possible that some of the moraines near the ice margin are ice-cored? Are any of these moraines stranded blue-ice moraines? If so, could subsequent ablation help explain some outliers? Bearing in mind the blue ice moraines at high altitudes in the TAM, it would be good to hear your view on this. 2. The start of the discussion is the place where you reference studies implying the presence of grounded ice in the Ross and its effect in blocking the flow of Transantarctic outlet glaciers. Later you make this an argument for the stability of the West Antarctic ice sheet for 15 Ma. Could you describe the evidence that the upper parts of Shackleton Glacier are affected by conditions near its convergence with Ross Sea ice? Once established for the reader, then the argument is strong. I was of the belief that there was little change higher up the transverse glaciers 3. lines 546-553.

Origin of debris from the base. Reference here the direct evidence of basal freezing near Mt Archernar? Eg. Bader et al, 2016, Q.S.R. and Graly et al. 2018, J.Glac. This seems more significant than reference to a general continental scale model. Ditto Uplift. Reference a fundamental paper on flexural uplift eg Stern & Tenbrink, JGR,1989, 94, p.10315? Technical corrections Fig 3 and caption.I found the labels on the Figure and the caption confusing. For example, where is B? And (b) seems to describe the highlighted area in A'. What does (c) show? Fig 8 and caption. Explain what 8c shows? Southwest Col drift not explicitly shown on the figure.

---

## Author Comment (AC1) · 1 Jun 2020

**Author response to referee comments on "A 14.5 million-year record of East Antarctic Ice Sheet fluctuations from the central Transantarctic Mountains, constrained with cosmogenic $_3$He, $_{10}$Be, $_{21}$Ne, and $_{26}$Al"**

We thank David Sugden for his review of the manuscript and his insightful comments.

Below, we address referee comments and describe additional, unsolicited changes that we've made to improve the manuscript. Referee comments are supplied in bold, with our responses in regular text.

**Specific comments:**

**1. Is it possible that some of the moraines near the ice margin are ice-cored? Are any of these moraines stranded blue- ice moraines? If so, could subsequent ablation help explain some outliers? Bearing in mind the blue ice moraines at high altitudes in the TAM, it would be good to hear your view on this.**

Although we recognize that the blue-ice moraine model is important at high elevation sites throughout the TAM, our field observations suggest that moraines at Roberts Massif are not stranded blue-ice moraines. In general, we characterize the moraines at Roberts Massif as boulder belt moraines, associated with thin drifts of angular boulders and directly overlying bedrock in several locations. Furthermore, the present ice fronts facing the moraine complexes are convex, have minimal debris within the ice, and do not have accumulating debris fields. These observations, coupled with the absence of modern blue-ice moraines at Roberts Massif, suggest that sediment supply to the glacier is low, which inconsistent with blue-ice moraines forming at other TAM locations today. While we do not interpret the Roberts Massif moraines as stranded blue-ice moraines, we acknowledge that (at least some) young outliers in our cosmogenic-nuclide dataset may result from ablation of a small ice core from moraines, and have added a sentence on this topic on lines 466-467 and a parenthetical reference on line 472.

**2. The start of the discussion is the place where you reference studies implying the presence of grounded ice in the Ross and its effect in blocking the flow of Transantarctic outlet glaciers. Later you make this an argument for the stability of the West Antarctic ice sheet for 15 Ma. Could you describe the evidence that the upper parts of Shackleton Glacier are affected by conditions near its convergence with Ross Sea ice? Once established for the reader, then the argument is strong. I was of the belief that there was little change higher up the transverse glaciers**

We agree that this argument needed clarification. Previous studies have shown that buttressing by Ross Sea ice affects ice thickness at the heads of TAM outlet glaciers, albeit significantly less thickening than at the mouths of these glaciers. To further our argument, we've updated the paragraph on lines 534-544 to include references that evidence the effect of buttressing ice in the Ross Sea on the uppermost reaches of TAM glaciers. Because the moraines at Roberts Massif mark times when the ice configuration was similar to today, we speculate that there was at least a buttressing ice shelf in the Ross Sea, or even a grounded ice sheet. Either of these scenarios would require inflow of ice to the Ross Sea from West Antarctica. However, we recognize that we cannot distinguish between a Ross Ice Shelf (ice configuration in the Ross Sea similar to today) and a Ross Ice Sheet (ice configuration similar to the Last Glacial Maximum) with our data. Therefore, we've also updated the paragraph on lines 534-544 and the final sentence of the paper on lines 690-693 to allow for either of these possibilities.

**3. Lines 546-553. Origin of debris from the base. Reference here the direct evidence of basal freezing near Mt Archernar? Eg. Bader et al, 2016, Q.S.R. and Graly et al. 2018, J.Glac. This seems more significant than reference to a general continental scale model.**

References to Bader et al. (2016) and Graly et al. (2018) added.

**4. Ditto Uplift. Reference a fundamental paper on flexural uplift eg Stern & Tenbrink, JGR,1989, 94, p.10315?**

Reference to Stern and ten Brink (1989) added.

**Technical corrections:**
**1. Fig 3 and caption. I found the labels on the Figure and the caption confusing. For example, where is B? And (b) seems to describe the highlighted area in A'. What does (c) show?**

We've clarified the inset labels in both the figure and the caption.

**2. Fig 8 and caption. Explain what 8c shows? Southwest Col drift not explicitly shown on the figure.**

We've added the word "drift" to the Southwest Col label in Figure 8a and 8b, and bolded it for clarity. We also added a description of inset c, which does not show the Southwest Col drift, to the figure caption.

**Additional changes to the manuscript:**

1. We correct a miscount in total sample numbers found in the original submission, which included samples from the Supplementary Information that are not critical to the interpretations discussed in the text. Thus, we revise the total number of samples discussed in the text from 180 to 168 on lines 13 and 70, as well as the breakdown of cosmogenic-nuclide measurements on lines 392-393.
2. Removed a stray '---' from line 489.
3. Changed erroneous section reference to Section 4.1 on line 529 to the correct Section 3.1.

---

## Referee Comment (RC2) · Julia Lindow (Referee) · 15 Jun 2020

**Significance and novelty:**

This paper focuses on the application of cosmogenic nuclide dating to ancient glacial deposits in Antarctica and shows how such information can be used to get a better understanding of paleo-ice sheet dynamics. The authors find that the East Antarctic Ice Sheet existed in similar to present extent since Miocene times (~15 Ma) near Shackleton Glacier. A pattern so far mostly seen in marine records off-shore East Antarctica,

and at the same time a topic of elevated interest for the community as the mid-Miocene period is considered to reflect climate conditions well within future predictions. Sampling and field work are outstanding in detail, quantity, and strategy. The analytical work has been conducted rigorously and with high scientific standards. Nuclide measurements are assessed thoroughly, and discussion of outliers/problems is suitable. Overall, data handling and presentation is transparent and comprehensible, and this paper offers 163 new exposure ages from over 20 moraines in a remote region in the central TAM. The combination of sample distribution, amount, and the measurement of up to four different cosmogenic nuclides (including two stable and two radiogenic) provides great detail regarding the samples (glacial) history and reduces the uncertainties that are part of dating of glacial deposits (e.g. nuclide inheritance, post-depositional disturbance). The authors present data from a large suite of moraines at Roberts Massif, central TAM, which shows increasing exposure ages with increasing distance and elevation away from the modern ice edge, a pattern one would expect from glacial thinning and/or rock uplift. In an area where terrestrial record is sparse this new data provides valuable evidence for an East Antarctic Ice Sheet configuration relatively stable since mid-Miocene with very low surface erosion (« 5 cm/Myr) and cold-based conditions for the East Antarctic Ice Sheet at least since 14.5 Ma. Isostatic rebound as consequence of glacial incision is found to be driving observed uplift of almost 200 m. The presented data does not exclude a reduced ice sheet during the mid-Pliocene warm period, as related evidence would be hidden beneath the current ice sheet. This is an important paper and it was a pleasure to read.

**Specific comments:**

1) Based on the detailed description of field work and sampling, the authors put a great deal of effort into sample selection and documentation, especially to minimize effects of common complications in surface exposure dating, e.g. nuclide inheritance or non-cosmogenic nuclides. So mainly out of curiosity, could some boulders of sufficient size have provided shielded samples to get direct measurements of inherited / non-

cosmogenic nuclides in combination with the surface samples?

2) No potential shielding from snow cover is discussed, and I assume it is considered negligible in respect to locality and the known average low snow accumulation. However, the age of the samples allows for some degree of uncertainty on seasonal or prolonged snow cover, and I would be interested to hear the authors thoughts on this.

3) Line 269-271: *"First described by Mercer (1972), the Sirius Group occurs throughout the upper (> ~2000 m elevation) TAM as erosional remnants of clay-rich diamicton that are correlated with at least one period of past temperate glaciation."* I read this as Sirius deposits are exclusively found above 2000 m, which could be misleading because there are Sirius Group outcrops are at lower elevations, e.g. Hambrey et al., 2003, and Mayewski 1975. I suggest changing the statement to > ~1500 m.

4) Line 571: *"≤ ~200m"*, this is a little odd, I would just write <~200 m

5) Section 4.1, Uplift at Roberts Massif: I understand the notion to compare potential uplift rates with existing data (here McMurdo Dry Valleys). However, I question the reliability of evaluating uplift rates or isostatic rebound over the extend of almost 1000 km, and thereby neglecting the influence of regional morphology and geologic structures. For me, the argumentation implies the whole TAM behaved as one block, undisturbed from north to south, while trough incision driven by glacial erosion (as discussed to be the main driver of uplift at Roberts Massif) can also (re-)activate underlying faults and induce block uplift (e.g. Studinger et al. 2006, or as shown for the Shackleton Range: Paxman et al., 2017). This would reflect in localized uplift rates which could be very different from the McMurdo Dry Valleys. I think this section would benefit from additional details on uplift along the TAM (e.g. Paxman et al., 2019).

**Technical corrections:**

Fig 1 and 3: missing scale bar and Lat/Lon labels (Fig 1), also, if possible, highlight/mark study area in figure 1

Fig 4, caption: no mention of (d) in the caption and missing reference to (d) under a); see : *"... with numbers corresponding to moraine names in (c) and letters A and A' corresponding to positions in (c)."*

Fig 5 (b), caption: It would be interesting to know the length of the pole for better scale or just give an approximate thickness.

Fig 11: text and axis labels are quite small, and rather hard to read

Fig 12: Please check numbering for BBY, BGE and WAL, it's different in figure 4. Also in the map (Fig 4) it is not quite clear which one is BGE.

Fig 14, caption: (c) is missing, and as a consequence subsequent description is off by one letter. *"Colors on the timescale at the bottom correspond to moraine colors in Figures 4, 7, and 8."* They don't, at least not for the reader, e.g. 'Pliocene' is more yellow then the orange of the moraines in the overview figures. Also, the color scheme used for the age data (d) implies a relation to the timescale used, which I find a little confusing. Maybe a different set of colors or symbols could make this figure clearer.

For the figures in general: The marker and information overlaying satellite maps are of mediocre quality/readability, which might be the result of compressing the images for this pre-print version, if not it would be worth looking into to ensure good quality images in the final version.

---

## Author Comment (AC2) · 19 Jun 2020

**Author response to referee comments on "A 14.5 million-year record of East Antarctic Ice Sheet fluctuations from the central Transantarctic Mountains, constrained with cosmogenic $_3$He, $_{10}$Be, $_{21}$Ne, and $_{26}$Al"**

We thank Julia Lindow for her thorough review of the manuscript and her insightful comments.

Below, we address referee comments and describe additional, unsolicited changes that we've made to improve the manuscript. Referee comments are supplied in bold, with our responses in regular text.

**Specific comments:**

**1. Based on the detailed description of field work and sampling, the authors put a great deal of effort into sample selection and documentation, especially to minimize effects of common complications in surface exposure dating, e.g. nuclide inheritance or non-cosmogenic nuclides. So mainly out of curiosity, could some boulders of sufficient size have provided shielded samples to get direct measurements of inherited / non- cosmogenic nuclides in combination with the surface samples?**

We did not collect samples shielded by larger boulders or from the undersides of boulders, although recognize that this technique may be of use for quantifying inherited and/or non-cosmogenic nuclides (e.g., Valletta et al., 2017). However, on lines 148–153 of the manuscript, we discuss previous estimates of non-cosmogenic $_3$He in Ferrar dolerite, noting that these values are within measurement error for our samples.

**2. No potential shielding from snow cover is discussed, and I assume it is considered negligible in respect to locality and the known average low snow accumulation. However, the age of the samples allows for some degree of uncertainty on seasonal or prolonged snow cover, and I would be interested to hear the authors thoughts on this.**

As this comment suggests, significant persistent snow cover is inconsistent with local climatology, extremely low subaerial erosion rates over the last 15 Ma, and salt accumulation in TAM soils. Further, the observation that Roberts Massif is a long-term ablation area, as evidenced by the surrounding modern blue-ice ablation zones and the abundant moraines (especially younger than 3 Ma) throughout the massif, supports the idea of low snow accumulation at this location. Given these observations, we've made the assumption that a snow cover correction is not necessary over the course of our record, despite the old age of the landforms.

**3. Line 269-271: "First described by Mercer (1972), the Sirius Group occurs throughout the upper (> ~2000 m elevation) TAM as erosional remnants of clay-rich diamicton that are correlated with at least one period of past temperate glaciation." I read this as Sirius deposits are exclusively found above 2000 m, which could be misleading because there are Sirius Group outcrops are at lower elevations, e.g. Hambrey et al., 2003, and Mayewski 1975. I suggest changing the statement to > ~1500 m.**

We've updated the text accordingly.

**4. Line 571: "≤~200m", this is a little odd, I would just write <~200 m.**

We've changed the text to read <~200 m.

**5. Section 4.1, Uplift at Roberts Massif: I understand the notion to compare potential uplift rates with existing data (here McMurdo Dry Valleys). However, I question the reliability of evaluating uplift rates or isostatic rebound over the extend of almost 1000 km, and thereby neglecting the influence of regional morphology and geologic structures. For me, the argumentation implies the whole TAM behaved as one block, undisturbed from north to south, while trough incision driven by glacial erosion (as discussed to be the main driver of uplift at Roberts Massif) can also (re-**

)activate underlying faults and induce block uplift (e.g. Studinger et al. 2006, or as shown for the Shackleton Range: Paxman et al., 2017). This would reflect in localized uplift rates which could be very different from the McMurdo Dry Valleys. I think this section would benefit from additional details on uplift along the TAM (e.g. Paxman et al., 2019).

We acknowledge the importance of the hypothesis that different TAM blocks have different uplift histories over the last 15 Ma. However, our data do not provide evidence for or against that hypothesis, but rather place bounds on the allowable amount of uplift at Roberts Massif over the course of our record. Given this, we don't discuss differential uplift across the TAM in this paper, but provide evidence for uplift rates elsewhere in the TAM for completeness.

**Technical corrections:**
**1. Fig 1 and 3: missing scale bar and Lat/Lon labels (Fig 1), also, if possible, highlight/ mark study area in figure 1.**

We've added a scale bar, lat/long labels, and a box highlighting the study area to for Figure 1. For Figure 3, we've included dimensions in the caption.

**2. Fig 4, caption: no mention of (d) in the caption and missing reference to (d) under a); see : "…with numbers corresponding to moraine names in (c) and letters A and A' corresponding to positions in (c)."**

We've added a description of and reference to panel (d) to the caption and further updated the caption for clarity.

**3. Fig 5 (b), caption: It would be interesting to know the length of the pole for better scale or just give an approximate thickness.**

The caption now includes the pole length (120 cm).

**4. Fig 11: text and axis labels are quite small, and rather hard to read.**

All font sizes will be revisited for final production files.

**5. Fig 12: Please check numbering for BBY, BGE and WAL, it's different in figure 4. Also in the map (Fig 4) it is not quite clear which one is BGE.**

The labels for BGE and WAL were erroneously switched in Figure 12. This has been corrected. Information about the NLO/NLI and POS moraine complexes, and clarification about the BBY and BGE moraines has also been added to the caption.

**6. Fig 14, caption: (c) is missing, and as a consequence subsequent description is off by one letter. "Colors on the timescale at the bottom correspond to moraine colors in Figures 4, 7, and 8." They don't, at least not for the reader, e.g. 'Pliocene' is more yellow then the orange of the moraines in the overview figures. Also, the color scheme used for the age data (d) implies a relation to the timescale used, which I find a little confusing. Maybe a different set of colors or symbols could make this figure clearer.**

We've corrected the panel references in the caption. To avoid confusion with the timescale color bar, we've removed these colors altogether as well as the reference to Figures 4, 7, and 8. Finally, we added a legend to panel d, rather than listing the colors in the figure caption, to improve clarity.

**7. For the figures in general: The marker and information overlaying satellite maps are of mediocre quality/readability, which might be the result of compressing the images for this pre-**

**print version, if not it would be worth looking into to ensure good quality images in the final version.**

We will ensure that the final version includes print-quality images.

**Additional changes to the manuscript:**

1. Numbers were switched for the WBK and POS moraines in Figure 4. This has been corrected.
2. Corrected erroneous section references on lines 509 and 524.
3. Corrected erroneous figure references on lines 505, 513, 519, 584, and 603.
4. Clarified figure reference location in sentence on lines 332–335.